# Detection of senescence using machine learning algorithms based on nuclear features

Imanol Duran [1,2], Joaquim Pombo[1,2], Bin Sun [1,2], Suchira Gallage [1,2,3,4], Hiromi Kudo[5], Domhnall McHugh[1,2], Laura Bousset [1,2], Jose Efren Barragan Avila [3], Roberta Forlano [6], Pinelopi Manousou[6], Mathias Heikenwalder[3,4,7], Dominic J. Withers [1,2], Santiago Vernia [1,2], Robert D. Goldin [5] & Jesús Gil [1,2] ✉

Cellular senescence is a stress response with broad pathophysiological implications. Senotherapies can induce senescence to treat cancer or eliminate senescent cells to ameliorate ageing and age-related pathologies. However, the success of senotherapies is limited by the lack of reliable ways to identify senescence. Here, we use nuclear morphology features of senescent cells to devise machine-learning classifiers that accurately predict senescence induced by diverse stressors in different cell types and tissues. As a proof-of-principle, we use these senescence classifiers to characterise senolytics and to screen for drugs that selectively induce senescence in cancer cells but not normal cells. Moreover, a tissue senescence score served to assess the efficacy of senolytic drugs and identified senescence in mouse models of liver cancer initiation, ageing, and fibrosis, and in patients with fatty liver disease. Thus, senescence classifiers can help to detect pathophysiological senescence and to discover and validate potential senotherapies.

Senescence is a cellular response that limits the replication of old, damaged, and cancerous cells. Senescent cells undergo a stable cell cycle arrest, produce a bioactive secretome (the senescence-associated secretory phenotype or SASP), and undergo many characteristic phenotypic changes[1]. Amongst those changes, senescent cells reprogram their metabolism, acquire a flat and enlarged morphology, display an increase in lysosomal mass[2], rearrange their chromatin[3–5], and undergo nuclear changes[6–9].

Senescent cells accumulate during aging, are present in cancerous and fibrotic lesions, and are often associated with disease[10]. Research in the last decade has shown that beyond these associations, lingering senescent cells contribute to aging and disease progression[11]. Consequently, there is growing interest in identifying drugs that selectively kill senescent cells, referred to as senolytics[12]. Clinical trials using senolytic drugs are still in their infancy[13,14] yet hold enormous potential, given the broad range of senescence-associated pathologies[10].

A key requirement for the success of senolytic clinical trials, and indeed to better understand senescence biology, is the reliable identification of senescent cells. Multiple markers to identify senescent cells exist, such as senescence-associated β-galactosidase (SA-β-Gal)

[1]MRC Laboratory of Medical Sciences (LMS), Du Cane Road, London W12 0NN, UK. [2]Institute of Clinical Sciences (ICS), Faculty of Medicine, Imperial College London, Du Cane Road, London W12 0NN, UK. [3]Division of Chronic Inflammation and Cancer, German Cancer Research Center (DKFZ), Heidelberg, Im Neuenheimer Feld 280, 69120 Heidelberg, Germany. [4]M3 Research Center for Malignome, Metabolome and Microbiome, Faculty of Medicine, University of Tübingen, Otfried-Müller-Straße 37, 72076 Tübingen, Germany. [5]Section for Pathology, Division of Digestive Diseases, Department of Metabolism, Digestion and Reproduction, Faculty of Medicine, Imperial College London, London W2 1NY, UK. [6]Liver Unit, Section of Hepatology and Gastroenterology, Division of Digestive Diseases, Department of Metabolism, Digestion and Reproduction, Faculty of Medicine, Imperial College London, London W2 1NY, UK. [7]Cluster of Excellence iFIT (EXC 2180), Eberhard Karls University, Tübingen, Germany. ✉e-mail: jesus.gil@imperial.ac.uk

activity[15], that reflects the increased lysosomal mass of senescent cells[2]. However, non-senescent cells such as macrophages often stain positive for SA-β-Gal, and SA-β-Gal can only be detected in vivo using cryosections, which complicates its use as a biomarker. Another widely used senescent marker is the cyclin-dependent kinase inhibitor p16[INK4a], which is induced as part of the senescence program to arrest cells[16]. However, p16[INK4a] is often deleted in cancer cells and it is difficult to detect p16[Ink4a] in mouse tissue sections with current antibodies. Therefore, due to a combination of technical issues and the complexity and heterogeneity of senescence, there is no such thing as universal senescence markers, and there is a need to rely on multiple markers in combination[1,17].

Recently, imaging-based approaches have been developed to identify senescence[18–20]. While these reports prove that image-based classifiers can identify senescent cells, to what extent such classifiers can be used easily by other labs, identify a variety of senescent cell types, or be applied to other contexts and questions, is unclear.

Here, we take advantage of nuclear features to devise machine-learning algorithms that identify senescence. We devise a family of algorithms that detect a wide range of senescent cells, from cancer cells to primary fibroblasts, with high accuracy. These algorithms require less computational power than image-based deep neural networks and can be adapted for use with data obtained from open-source image analysis software, which facilitates their use by others. Finally, we took advantage of our nuclear feature-based senescence classifiers to identify drugs that selectively induce senescence in cancer cells and to monitor senescence in different mouse models and patient samples. In summary, these senescence classifiers can help to elucidate the pathophysiological roles of senescence and assist in the discovery and validation of senotherapies as well as to better stratify patients.

## Results

### Nuclear features can be used to devise classifiers that predict senescence in human cells

To identify nuclear features that can be used to detect senescent cells, we induced senescence in A549 human lung adenocarcinoma cells by treating them with the chemotherapeutic agent etoposide, a topoisomerase II inhibitor (Fig. 1a). In contrast with DMSO-treated cells, a high percentage of etoposide-treated cells stained positive for senescence-associated-β-galactosidase (SA-β-Gal) (Fig. 1b). Moreover, most etoposide-treated cells were negative for BrdU staining, indicating cell cycle arrest, had upregulated DNA damage (as assessed by γH2AX staining), and increased levels of p53 and its target the cyclin-dependent kinase inhibitor p21[CIP1], consistent with senescence (Supplementary Fig. 1a–d). To further examine senescence induction, we took advantage of quantitative immunofluorescence (IF) and searched for cells expressing a combination of those markers (such as cells that were BrdU negative and p21[CIP1] positive, Fig. 1c and Supplementary Fig. 1e; double p53 /p21[CIP1] positive, Fig. 1d and Supplementary Fig. 1f; and double p53 /γH2AX positive, Supplementary Fig. 1g). The above analysis shows that most of the etoposide-treated A549 cells were senescent.

Senescent cells are known to change chromatin architecture and nuclear morphology[4,8,9]. To assess the nuclear morphology of senescent and control A549 cells, we stained the nuclei with 4′,6-diamidino-2-phenylindole (DAPI) and used a high-throughput automated microscopy system (IN Cell Analyzer 2500HS). Although nuclear morphology and size were heterogeneous, a high proportion of senescent A549 cells had bigger nuclei that were morphologically distinct from those of control cells (Fig. 1e). We used image analysis software (IN Carta) to examine individual nuclear features, including nuclear area, gyration radius, compactness, chord ratio, displacement, elongation, and form factor. All of these nuclear features, except form factor, were significantly different between senescent

(etoposide-treated) and control (DMSO-treated) A549 cells (Fig. 1f and Supplementary Fig. 2a).

As none of these nuclear features alone could distinguish senescent cells from non-senescent cells, we used these features to devise machine-learning classifiers that could predict senescence. To this end, we developed datasets to train random forest and classification tree-based machine-learning algorithms (Supplementary Fig. 2b, c and Supplementary Table 1). Initially, we generated two senescence classification models (Supplementary Table 2), termed AEM (classification tree-based) and AERFM (random forest-based). Given that the majority of the etoposide-treated cells were senescent (Supplementary Table 3) and conversely the majority of the DMSO-treated cells were not senescence, to simplify the training of these algorithms, we assumed that all the etoposide-treated cells were senescent and all the DMSO-treated cells were non-senescent (Fig. 1g). Analysis of the training sets showed that both classifiers identified senescence in etoposide-treated A549 cells to a similar extent as staining for SA-β-Galactosidase activity (Fig. 1h), which we validated with test data from new samples (Fig. 1i). Analysis of precision-recall and receiver operating characteristic (ROC) curves, which represent sensitivity as a function of fall-out, showed high specificity in the detection of senescent cells by both the AEM and AERFM classifiers (Supplementary Fig. 2d, e).

To explore whether this approach could be widely adopted, we took advantage of CellProfiler[21], an open-source image analysis software. CellProfiler allowed us to examine 17 nuclear features. Like what we observed with In Carta, most nuclear features were significantly different in senescent cells (Supplementary Fig. 3a). AEMCP (classification tree-based) and AERFMCP (random forest-based) classifiers trained using datasets produced with CellProfiler, identified senescence in both training and test datasets of etoposide-treated A549 cells to a similar extent as SA-β-Galactosidase immunostaining (Supplementary Fig. 3b, c). Quality parameters further validated the classifiers generated with CellProfiler data (Supplementary Fig. 3d, e). Overall, we infer that machine-learning algorithms based on nuclear features can identify senescent cells.

### Predictors can distinguish senescent cells from those undergoing quiescence or DNA damage

One of the defining characteristics of senescent cells is their stable cell cycle arrest[1,17]. Because quiescent cells are also arrested in the cell cycle, we wanted to determine whether our classifiers can distinguish senescent cells from quiescent cells. To this end, we compared A549 cells that were growing (10% FBS), quiescent (0.5% FBS), and senescent (treated with 2 μM etoposide, Fig. 2a). BrdU incorporation confirmed that a significant proportion of A549 cells cultured in 10% FBS were dividing, whereas most cells cultured in 0.5% FBS or treated with etoposide were arrested (Fig. 2b, c). In contrast, only etoposide-treated A549 cells were senescent, as shown by SA-β-Gal staining (Fig. 2b, d). As expected, both the AEM and AERFM classifiers predicted a significant proportion of senescent cells in the etoposide-treated cultures (Fig. 2e and Supplementary Fig. 4a). Importantly, A549 cells cultured in 0.5% FBS were not classified as senescent by our algorithms, even though most cells were arrested (Supplementary Fig. 4b). Overall, these results indicate that our senescence classifiers can accurately distinguish senescent cells not only from growing but also from quiescent cells.

Since cell confluency can affect proliferation rates and be a confounding factor when staining for SA-β-Gal activity[22], we assessed our classifier on proliferating or senescent cells seeded at different densities (Fig. 2f). These experiments show that our classifier was able to accurately identify senescent cells regardless of confluency (Fig. 2g, h and Supplementary Fig. 4c).

DNA damage occurs in most types of senescence and is a key driver of senescent phenotypes[23]. To understand if the nuclear changes detected by our algorithm reflected alterations caused by a DNA damage response rather than senescence, we compared how the AEM

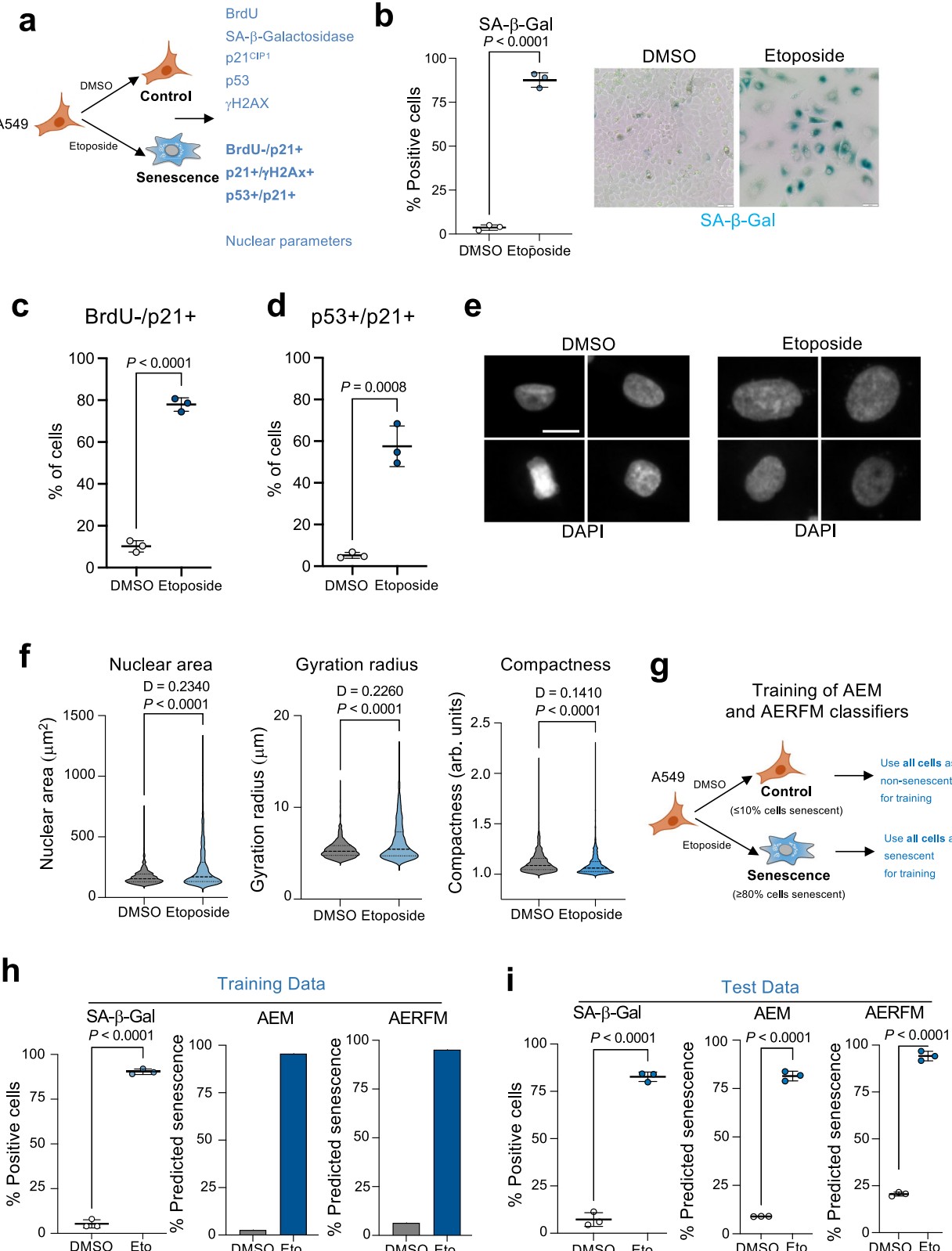

classifier identified cells undergoing DNA damage (DD, induced after 36 h of γ-irradiation) or senescence caused by the aurora A kinase inhibitor MLN8054[24] or etoposide (Fig. 2i). While A549 cells undergoing DD or senescence with MLN8054 or etoposide were arrested and upregulated p21[CIP1] expression (Supplementary Fig. 4d, e), a significant increase in cells showing 53BP1 foci in their nuclei, indicative of DNA damage was only observed in irradiated cells or cells in which

senescence was induced with etoposide (Fig. 2j and Supplementary Fig. 4f). SA-β-Gal staining confirmed that MLN8054- and etoposide-treated, but not cells irradiated for 36 h were senescent (Fig. 2k). Finally, the AEM classifier identified most etoposide-treated A549 cells as senescent. It also identified a proportion (less than 30%) of the irradiated cells as senescent, suggesting that changes induced by DNA damage (that are central to most types of senescence) might be

**Fig. 1 | Nuclear features can be used to identify senescent cells. a** Experimental design for the induction and characterization of senescence in A549 cells treated with 2 μM etoposide for 7 days. **b** Quantification of SA-β-galactosidase (SA-β-Gal) activity after DMSO or etoposide treatment ($n = 3$, left) and representative images of SA-β-Gal staining (right). Scale bar, 50 μm. **c, d** Quantification of Brdu-/p21$^{Cip1}$+ cells (**c**) and p21/p53 double-positive cells (**d**) treated with DMSO or etoposide ($n = 3$). **e** Representative images of DAPI-stained nuclei for DMSO-treated (normal) and etoposide-treated (senescent) A549 cells. Scale bar, 20 μm. **f** Quantification of nuclear features for DMSO and etoposide-treated A549 cells 7 days post-treatment ($n = 1000$ cells per group). Kolmogorov–Smirnov test was performed to assess probability distribution, with D value indicated. Data of a representative experiment out of 3. **g** Experimental design for the development of training sets for the AEM and AERFM senescence classifiers. **h** Analysis of training datasets of A549 cells treated with DMSO or etoposide. Percentage of SA-β-Gal positive cells (left, $n = 3$) and percentage of predicted senescent cells using the AEM (center) and AERFM (right) senescence classifiers. **i** Analysis of test datasets of A549 cells treated with DMSO or etoposide. Percentage of SA-β-Gal positive cells by immunofluorescent C$_{12}$FDG in DMSO (normal) and etoposide-treated A549 cells (senescent) (left, $n = 3$). Percentage of predicted senescent cells in the validation datasets using the AEM (middle) and AERFM (right) classifiers. AEM and AERFM prediction was performed on the same cells as SA β-Gal staining. Error bars represent mean ± s.d.; $n$ represents number of replicates except when indicated differently. All statistical significances unless indicated differently were calculated using unpaired, two-tailed, Student's $t$-tests. AEM, A549 etoposide model; AERFM, A549 etoposide random forest model. Source Data are provided in the Source Data File.

detected by our classifiers. However, the fact that the AEM classifier identified MLN8054 cells (Fig. 2l, in which we did not observe significant amounts of DNA damage) as senescent, shows that the classifier is not just detecting DNA damage-associated changes but rather a combination of nuclear features associated with senescent cells.

## Predictors identify senescent cells with high accuracy and sensitivity

To better evaluate the predictive value of our classifiers, we examined mixtures of different ratios of non-senescent (DMSO-treated) and senescent (etoposide-treated) A549 cells (Fig. 3a, explained in Supplementary Fig. 5). We treated cells with DAPI, which stains DNA and identifies nuclei, and with 5-dodecanoylaminofluorescein di-β-D-galactopyranoside (C$_{12}$FDG), which is cleaved by β-galactosidase to generate a fluorescent product; this allowed us to simultaneously evaluate nuclear morphology parameters and SA-β-Gal activity, respectively (Fig. 3b). We established a fluorescence intensity cut-off for the identification of SA-β-Gal positive, senescent cells (Supplementary Fig. 6a, b).

The percentage of cells predicted to be senescent by both the AEM and AERFM classifiers significantly correlated with the percentage of SA-β-Gal positive cells (Fig. 3c and Supplementary Fig. 6c). Co-culture experiments also showed that predictions from the AEM classifier significantly correlated with the percentage of senescent cells as defined by considering p21$^{CIP1}$ positive cells (Supplementary Fig. 6d), p21$^{CIP1}$ positive/BrdU negative cells (Fig. 3d) or p21$^{CIP1}$/p53 double-positive cells (Fig. 3e).

To generate and train those classifiers, we assumed that all etoposide-treated cells were senescent and all the DMSO-treated cells were normal (Fig. 1g). To understand if we could improve our algorithms by training them with cells that we have identified as senescent (or non-senescent) based on senescent markers, we generated a new classifier (that we called BAEM). We trained this classifier with senescent cells (from etoposide-treated cultures) that were SA-β-galactosidase positive, and non-senescent cells (from DMSO-treated cultures) that were negative for SA-β-galactosidase staining (Fig. 3f). We took a similar approach to generate new algorithms trained with p21$^{CIP1}$/p53 double-positive cells (PPEM, Supplementary Fig. 6e, f) or with BrdU negative/ p21$^{CIP1}$ positive cells (BPEM, Supplementary Fig. 6g, h). In all cases, the percentage of cells predicted to be senescent by the BAEM, PPEM, and BPEM classifiers significantly correlated with the percentage of SA-β-Gal positive cells in mixed cultures (Fig. 3g and Supplementary Fig. 6f, h). To confirm that these results were not biased by the time at which we co-cultured the normal and senescent cells in our protocol (a day before assessing senescence), we conducted experiments in which normal and senescent cells were co-cultured 4 days before the assessment of senescence, obtaining similar results (Supplementary Fig. 7).

To directly compare the assumption-based and marker-based classifiers, we employ them to predict senescence in the same dataset

(Fig. 3h). We assessed the precision (the ratio of predicted senescent cells to SA-β-Gal positive cells), accuracy (the ratio of true prediction, both for senescence and non-senescent cells), and recall (how many of the true senescent cells were identified by the classifier) and F$_1$ score for the predictions (Fig. 3i). This showed us that the marker-based algorithms had similar levels of performance to the AEM algorithm (where we had assumed that all the etoposide-treated cells were senescent), suggesting that training with cell populations comprised mostly of senescent cells is sufficient to generate efficient senescent classifiers.

## A family of machine-learning algorithms accurately predict senescence

To examine the generality of the senescence classifiers, we established co-cultures with different ratios of non-senescent A549 cells (DMSO-treated) and A549 cells induced to senesce by treatment with doxorubicin or the aurora kinase inhibitor barasertib (Supplementary Fig. 8). Additionally, we established mixed co-cultures of other human cell types treated with DMSO or etoposide, including a liver adenocarcinoma cell line (SK-HEP-1), melanoma cell line (SK-MEL-103), breast cancer cell line (MCF7) and colon cancer cell line (HCT116, Supplementary Fig. 9) or human fibroblasts (IMR90 cells, Supplementary Fig. 10). We stained cells with C$_{12}$FDG and DAPI to evaluate senescence and nuclear parameters respectively and used the senescence classifiers to predict senescence. The predictions and experimental data correlated significantly and very highly for the different cell types treated with etoposide (Supplementary Figs 9, 10). For most conditions, the precision, accuracy, and recall of both classifiers were comparable to etoposide-treated, senescent A549 cells (Fig. 4 and Supplementary Fig. 11).

Next, we generated new classifiers trained with data of senescence induced in different cell types (Fig. 4a and Supplementary Tables 2, 3). Five of the classifiers were generated using decision tree algorithms (GM, AEM, MEM, MERFM, HERFM), three were generated using random forest algorithms (AERFM, MERFM, HERFM) and we also included a voting-based consensus algorithm (VCA), whose decisions were based on the consensus of the other eight classifiers (Fig. 4a). Seven of the classifiers were trained using data from individual models of senescence, but the general model (GM) classifier was trained with data from 12 different senescence conditions (Supplementary Table 1). We observed a significant, high correlation between the percentage of SA-β-Gal positive cells and the percentage of cells predicted to be senescent in our nine co-culture datasets (Fig. 4b), although barasertib-treated A549 cells analyzed with GM, AEM, or IEM displayed a lower but still significant correlation. The accuracy and precision of most predictions were relatively high, particularly for the GM, AEM, and IEM classifiers (Fig. 4c and Supplementary Fig. 11a), whereas the random forest-based-AERFM and the voting VCA classifiers had the best recall rates (Supplementary Fig. 11b). Overall, looking at the F$_1$ score (Fig. 4d) and other parameters, the GM classifier and the VCA

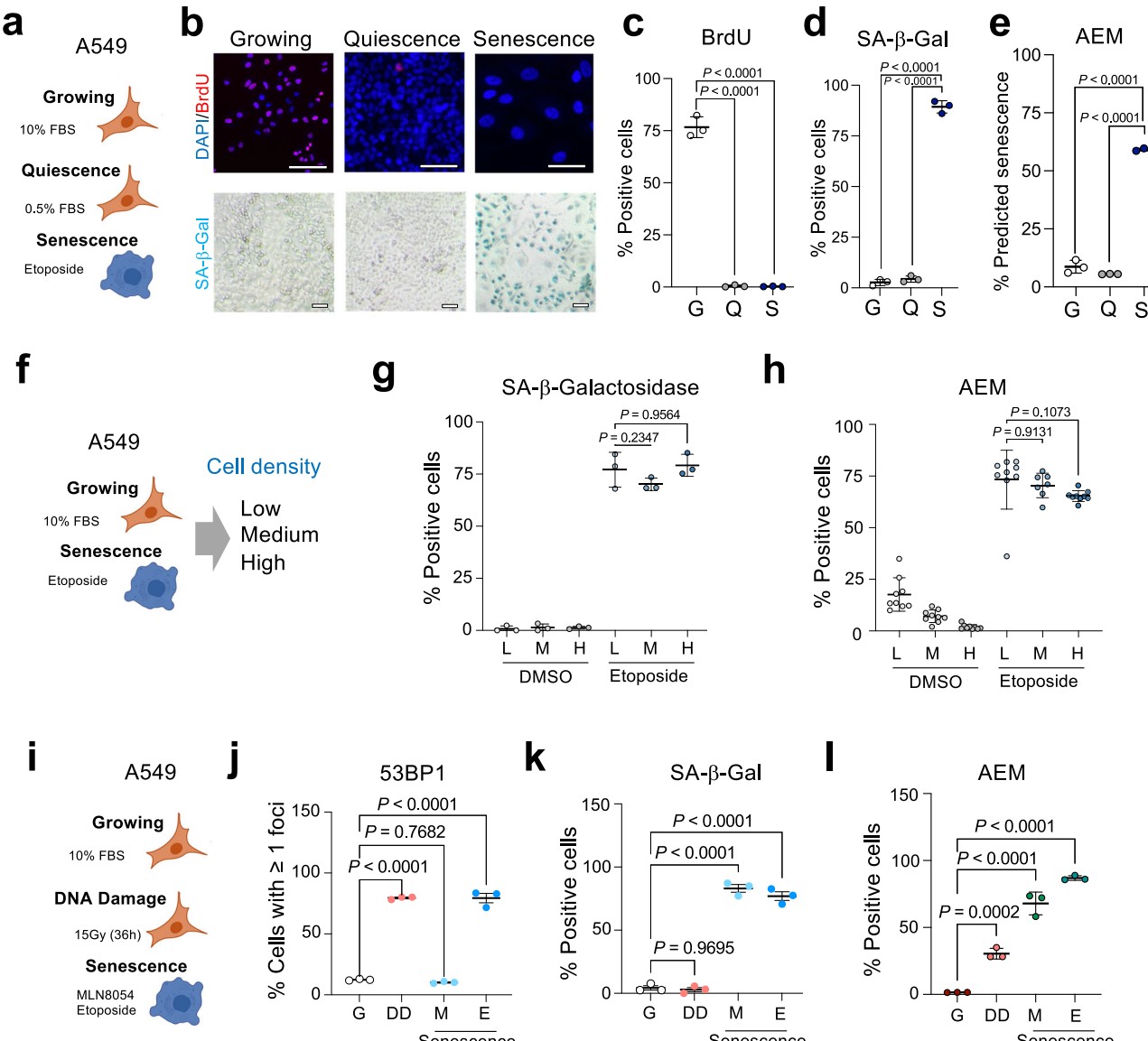

**Fig. 2 | Senescence classifiers distinguish senescent cells from those undergoing quiescence or DNA damage. a** Experimental design for the comparison of growing, quiescent and senescent A549 cells. **b** Representative images of growing (10% FBS), quiescent (0.5% FBS), and senescent (10% FBS, etoposide-treated) A549 cells stained with antibodies recognizing BrdU (top). SA-β-galactosidase (SA-β-Gal) staining from the same experiment is shown at the bottom. Scale bars, 100 μm. **c**, **d** Percentage of cells incorporating BrdU (**c**) and staining positive for SA-β-Gal (**d**) (*n* = 3). **e** Percentage of cells predicted to be senescent by the AEM senescence classifier (*n* = 3). **f** Experimental design for assessing the impact of cell density in senescent prediction. **g** Percentage of SA-β-Gal positive A549 cells treated with either DMSO or etoposide and seeded at low (L), medium (M), and high (H) density

(*n* = 3 per condition). **h** Senescence prediction using the AEM classifier of cultures of A549 cells treated with either DMSO or etoposide and seeded at low (L), medium (M), and high (H) density (*n* = 9 for all cases except etoposide-treated cells seeded at medium density, *n* = 7). **i** Experimental design for assessing the impact of DNA damage in senescence prediction. **j** 53BP1 positive cells in cultures of growing (G), irradiated (DD), and senescent (MLN8054 M; etoposide E) cells (*n* = 3). **k**, **l** Percentage of SA-β-Gal positive (**k**) and AEM positive (**l**) cells for the different conditions (*n* = 3). Error bars represent mean ± s.d.; *n* represents independent experiments. Statistical significance was calculated using multiple comparisons one-way ANOVA. Source Data are provided in the Source Data File.

classifier both performed well and consistently across datasets. Thus, most of the classifiers tested are suitable for predicting different types of senescence.

## Predictors can assist in the characterization and discovery of senotherapies

Drugs such as the BCL-2 family inhibitors ABT-263 and ABT-737 selectively kill senescent cells and have the potential to treat a wide range of diseases in which senescent cells accumulate[25–27]. To understand if our senescence classifiers could assist in the assessment of senolytic drugs, we setup a co-culture assay using A549 GFP

senescent cells (that have been treated with etoposide) and control A549 mCherry cells (treated with DMSO). In this way, we could assess the relative numbers of senescent and non-senescent cells by identifying the cells as GFP or Cherry positive (Fig. 5a and Supplementary Fig. 12). We treated the co-cultures with DMSO (as a control) or with two senolytic drugs (1 μM ABT-263 or 5 μM ABT-737) and analyzed the effects after 72 h. While non-senescent cells (A549 Cherry) increased in numbers during this period (Fig. 5b, c), senescent cells were selectively reduced upon treatment with both senolytic drugs, as assessed by counting A549 GFP positive cells (Fig. 5d) or using the AEM senescence classifier (Fig. 5e). This experiment

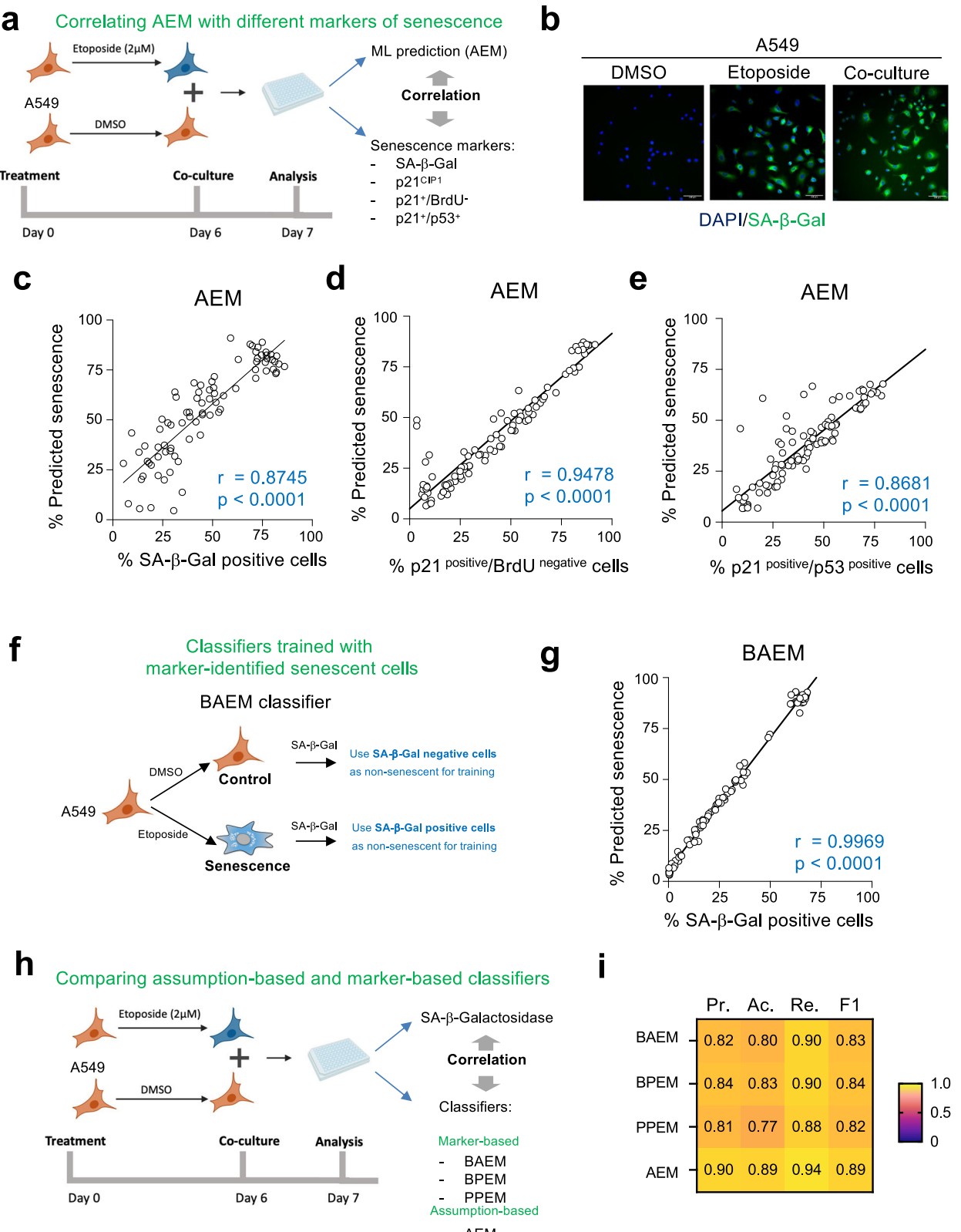

**a** Correlating AEM with different markers of senescence

**b** A549

DAPI/SA-β-Gal

**c** AEM — r = 0.8745, p < 0.0001

**d** AEM — r = 0.9478, p < 0.0001

**e** AEM — r = 0.8681, p < 0.0001

**f** Classifiers trained with marker-identified senescent cells

**g** BAEM — r = 0.9969, p < 0.0001

**h** Comparing assumption-based and marker-based classifiers

**i**

|      | Pr.  | Ac.  | Re.  | F1   |
|------|------|------|------|------|
| BAEM | 0.82 | 0.80 | 0.90 | 0.83 |
| BPEM | 0.84 | 0.83 | 0.90 | 0.84 |
| PPEM | 0.81 | 0.77 | 0.88 | 0.82 |
| AEM  | 0.90 | 0.89 | 0.94 | 0.89 |

shows how we could use senescent classifiers to characterize senolytic drugs.

Chemotherapies must induce senescence in the tumor to achieve a therapeutic outcome[28]. However, many side effects of chemotherapy arise from off-target induction of senescence in healthy, uninvolved tissues[29]. Some efforts to identify drugs inducing senescence of cancer cells have been successful[30], but screening for such drugs relies on having trustworthy readouts for senescence in cancer cells that are amenable to high-throughput screening. We reasoned that senescence classifiers could serve as the basis of such a screen.

As a proof of principle, we screened a collection of 676 drugs (selected from the Selleck Target Selective and Protein Kinase Inhibitor Library II libraries) for their ability to induce senescence in A549 cancer cells but not in IMR90 normal human fibroblasts (Fig. 5f). As

**Fig. 3 | Classifiers identify senescence at the single-cell level. a** Design of the experiments analysing co-cultures of DMSO-treated (normal) or etoposide-treated (senescent) A549 cells. **b** Representative pictures out of three independent experiments of cells stained for SA-β-Galactosidase (SA-β-Gal) activity using $C_{12}$FDG. Scale bars, 100 μM. **c–e** Correlation between percentage of SA-β-Gal positive (**c**) ($r = 0.8745$; $p < 0.0001$), p21$^{CIP1}$+/BrdU- (**d**) ($r = 0.9478$; $p < 0.0001$) and p21$^{CIP1}$+/p53+ (**e**) ($r = 0.8681$; $p < 0.0001$) cells and percentage of cells predicted to be senescent using AEM. Pearson correlation coefficient (two-tailed, 95% CI). r represents two-tailed nonparametric correlation probability. $n = 70–96$ wells. Wells are co-cultures of senescence and non-senescent cells at different ratios as explained in methods. **f** Experimental design for the selection of cells to train the BAEM classifier. **g** Correlation between percentage of SA-β-Gal positive cells and

cells predicted to be senescent using BAEM ($r = 0.9969$; $p < 0.0001$); Pearson correlation coefficient (two-tailed, 95% CI). **h** Experimental design to compare and validate assumption-based and marker-based classifiers by control (DMSO) and senescent (etoposide) cell co-culture and corresponding SA-β-Gal staining. **i** Heatmap of precision (Pr.), accuracy (Ac.), recall (Re.), and $F_1$ score (F1) median values (left to right) of BAEM, BPEM, PPEM, and AEM classifiers (top to bottom). Measures represent median values, calculated from $n = 70–96$ wells each. Wells are co-cultures of senescence and non-senescent cells at different ratios as explained in methods. BAEM SA-β-Gal+ A549 etoposide model, BPEM Brdu-/p21$^{CIP1}$+ etoposide model, PPEM p53+/p21$^{CIP1}$+ etoposide model, AEM. Source Data are provided in the Source Data File.

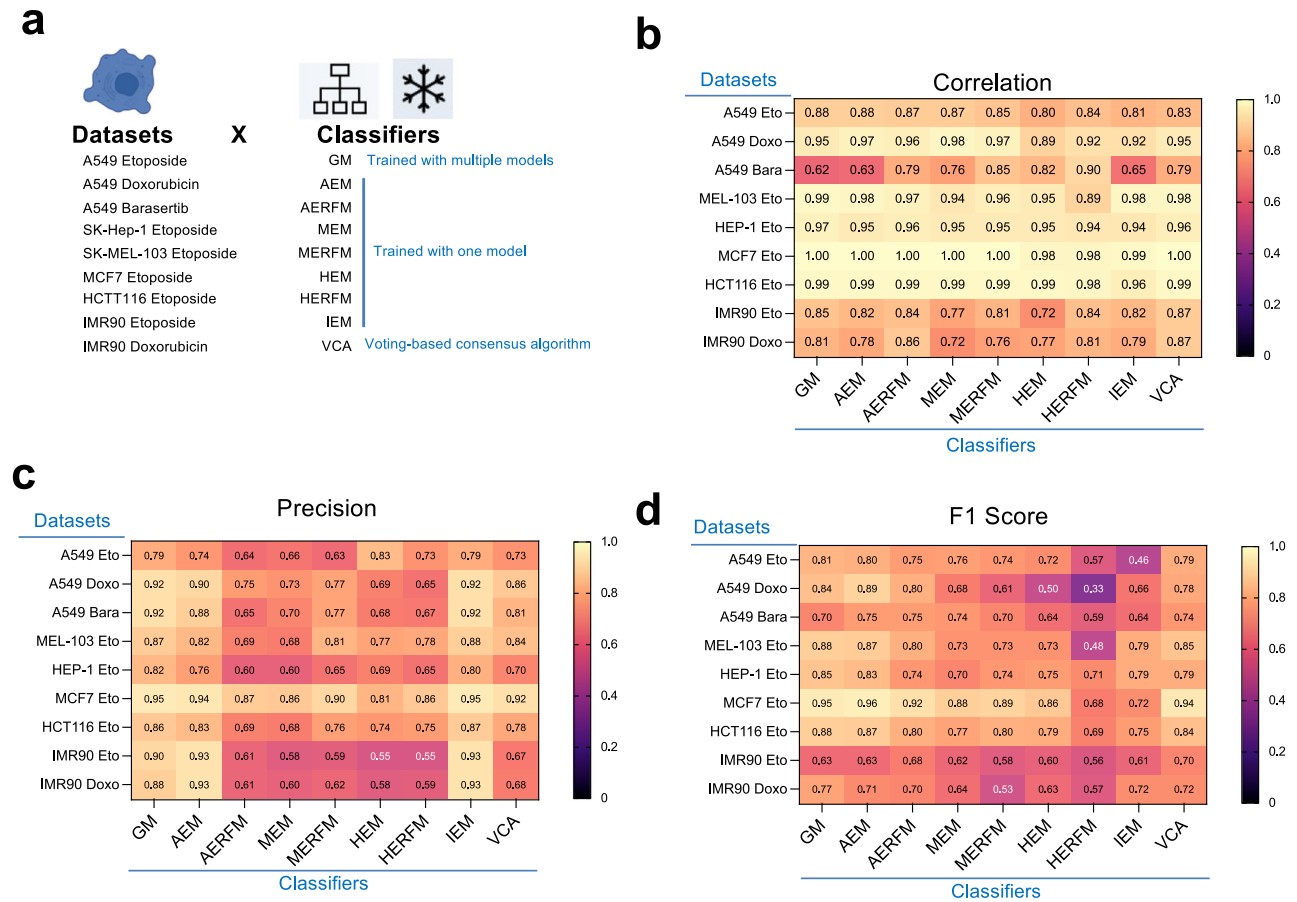

**Fig. 4 | Comparison of the performance of different senescence classifiers. a** Experimental design for fitness assessment of different senescence classifiers in datasets derived from co-cultures of senescent cells with non-senescent counterparts. **b–d** Correlation (**b**), precision (**c**), and $F_1$ score (**d**) of senescence classifiers (x-axis) in the different datasets (derived from co-cultures of different types of senescent cells with non-senescent counterparts) represented in heatmaps.

Measures represent median values, calculated from $n = 70–96$ wells each. Wells are co-cultures of senescence and non-senescent cells at different ratios as explained in methods. Each heatmap contains 9 algorithms (GM, AEM, AERFM, MEM, MERFM, HEM, HERFM, and IEM) and the last column corresponds to a voting-based consensus algorithm (VCA), obtained by equal weight voting system of the previous 8 algorithms. Source Data are provided in the Source Data File.

controls of the screen, we included doxorubicin, which induced senescence in both A549 and IMR90 cells. We excluded from the analysis drugs that were toxic (killing more than 60% of cells when compared with the controls). Using the GM classifier, we identified 56 drugs that were predicted to induce senescence in either A549 and/or IMR90 cells (Fig. 5g). Amongst those drugs 27 (exemplified by the aurora kinase inhibitors MLN8054) induced senescence only in A549 cells, 11 (e.g., the PARP1 inhibitor AG-14361) induced senescence only in IMR90 cells and 18 drugs induced senescence in both cells (Fig. 5h–j). Overall, the above experiments show how our senescence classifiers can assist in identifying different senotherapies.

## Characterization of drugs that induce senescence specifically in cancer cells

To further characterize if the drugs identified in our screen selectively induced senescent in A549 cells, we selected a few candidates, including the aurora kinase inhibitors MLN8054, and ZM447439, the Eg5 inhibitor ARQ621 and the gp130 inhibitor SC144. In addition, we treat cells with AG-14361 as an example of a drug inducing senescence selectively in IMR90 cells and doxorubicin and niraparib, drugs inducing senescence in both cell types. SA-β-Gal staining confirmed that our 4 candidates (MLN8054, ZM447439, ARQ621, and SC144) induced senescence in a higher percentage of A549 cells than IMR90 cells, whereas AG-14361 had the opposite effect (Fig. 6a, b).

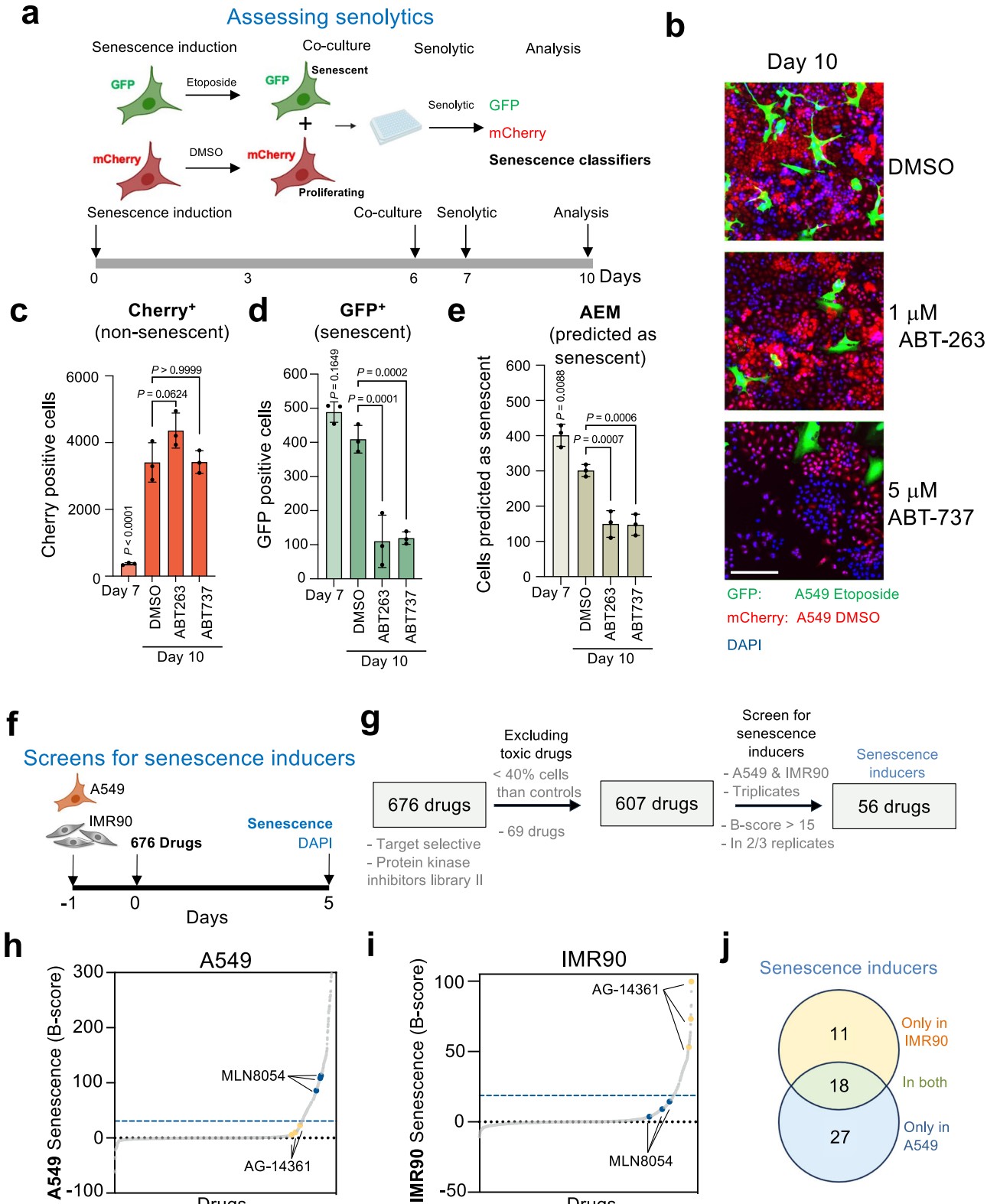

**Fig. 5 | Senescence classifiers can assist in characterizing and identifying senotherapies.** **a** Experimental design and timeline for assessing senolytic drugs. **b** Representative images of co-cultures of senescent (GFP) and non-senescent (mCherry) A549 cells 72 h after treatment with DMSO (top), ABT-263 (middle), and ABT-737 (bottom). Scale bar, 200 μM. **c–e** Quantification of mCherry positive cells (**c**), GFP positive cells (**d**) and cells predicted to be senescent by AEM (**e**), before and after treatment with the different drugs (*n* = 3). **f** Experimental design, and timeline for the screen for drugs inducing senescence. Drugs were used at 10 μM. **g** Summary of the screen results. **h, i** Distribution of B-score results in A549 (**h**) and IMR90 (**i**). The blue dashed line indicates the threshold (B-score > 15). Examples of an A549-specific hit MLN8054 (blue) and an IMR90-specific AG-14361 (yellow) are highlighted. **j** Venn Diagram for senescence-inducing drugs with selectivity for IMR90 (yellow), A549 (blue) or both (green). Data represent mean ± s.d. *n* represents independent experiments. Source Data are provided in the Source Data File.

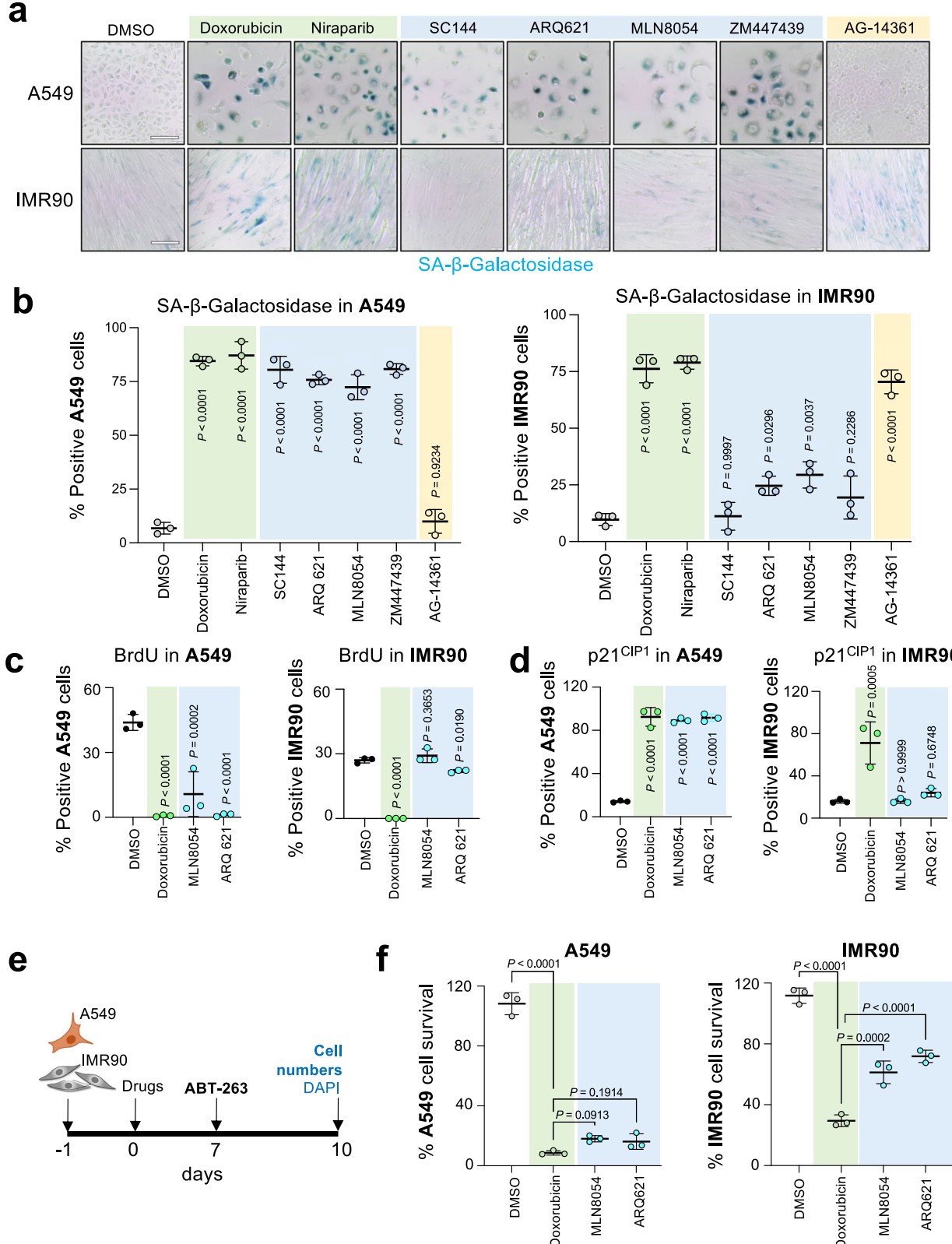

**Fig. 6 | Characterization of MLN8054 and ARQ621 as drugs preferentially inducing senescence in A549 over IMR90 cells. a** Images of SA-β-galactosidase (SA-β-Gal) staining on A549 (top) and IMR90 (bottom) cells treated with the indicated drugs. Representative images of one out of 3 experiments. Scale bar, 100 μM. **b** Percentage of SA-β-Gal positive cells after 7-day treatment with the indicated drugs in A549 cells (left) and IMR90 (right) cells ($n = 3$). **c, d** Quantification of BrdU (**c**) and p21$^{CIP1}$ (**d**) positive cells by immunofluorescence in A549 (left) and IMR90 (right) cells after treatment with selected drugs ($n = 3$). **e** Experimental design and timeline of the senolytic experiment quantified in (**f**). **f** Senolytic activity after treatment with 1 μM ABT-263 for 72 h in either A549 (left) or IMR90 (right) cells treated for 7 days with the indicated drugs ($n = 3$). Drug concentrations used in (**c, d**), **f** doxorubicin, 0.5 μM; MLN8054, 3 μM; ARQ 621, 0.1 μM. Significance was calculated with a one-way ANOVA (Tukey's multiple comparisons test). Data represent mean ± s.d. $n$ represents independent experiments. Source Data are provided in the Source Data File.

We decided to characterize in more detail the effects of ARQ621, an Eg5 inhibitor that has not been previously linked with senescence, and MLN8054, an aurora A kinase inhibitor previously shown to induce senescent in tumor cells[24]. Treatment with doxorubicin caused cell cycle arrest (Fig. 6c) and induced p53 and p21$^{CIP1}$ in both IMR90 and A549 cells (Fig. 6d and Supplementary Fig. 13a, b). While CDKN2A (encoding for p16$^{INK4a}$) is mutated in A549 cells[31], we also observed an upregulation of CDKN2A transcripts in IMR90 cells upon treatment with doxorubicin (Supplementary Fig. 13c). In contrast, treatment with MLN8054 or ARQ621 caused a significant cell cycle arrest, induction of p53 and p21$^{CIP1}$, and upregulation of SASP components in A549 but not in IMR90 cells (Fig. 6c, d and Supplementary Fig. 13), consistent with their differential effects on senescence induction.

Senolytic drugs such as ABT-263 can be used in combination with senescence-inducing drugs to target cancer cells, in what is termed the 'one-two punch' approach[32]. To determine if the identified compounds can be combined with senolytics to selectively kill cancer cells, we pre-treated A549 and IMR90 cells with these drugs for 7 days followed by treatment with ABT-263 for 3 days (Fig. 6e). Pre-treatment with doxorubicin, which induced senescence in both cell lines, also sensitized both IMR90 and A549 cells to ABT-263 (Fig. 6f). In contrast, pre-treatment with MLN8054 or ARQ621 sensitized over 75% of A549 cells to ABT-263 treatment, whereas less than half of the IMR90 cells became sensitive to ABT-263 (Fig. 6f). Overall, we conclude that our senescence classifiers can be used to discover drugs with selectivity to induce senescence in cancer over non-oncogenic cells.

## Predictors inform a tissue senescence score to detect senescence during cancer initiation in vivo

Next, we aimed to adapt our strategy to detect senescence in tissue sections. We took advantage of a model of liver pre-neoplasia and tumor initiation (Fig. 7a). Briefly, transposon-mediated transfer of oncogenic NRas (NRas$^{G12V}$) in hepatocytes is known to induce senescence, as inferred from elevated levels of p16$^{Ink4a}$ and p21$^{Cip1}$, elevated SA-β-galactosidase activity and a senescence-associated secretory phenotype[33–35]. We confirmed a significant increase in different senescent markers, including p21$^{Cip1}$, the SASP components uPAR[36], and the open reading frame 1 (ORF1) product of the LINE1 transposon[37], specifically in NRAS-positive cells of mice transduced with a transposon expressing oncogenic NRas$^{G12V}$ but not on those expressing an NRas$^{G12V, D38A}$ inactive mutant (Fig. 7b, c and Supplementary Fig. 14a, b).

Taking advantage of this system, we detected senescent cells in NRas$^{G12V}$-induced preneoplastic liver sections by performing immunohistochemistry for p21$^{Cip1}$ (Supplementary Fig. 14c, d). We then used the slide image analysis software QuPath to measure different nuclear features in p21-positive and p21-negative populations. p21-positive cells displayed generally enlarged nuclei, with bigger areas, more extreme caliper values (also known as Feret diameter), and reduced circularity (Supplementary Fig. 14e) as observed with senescent cells in culture. Using a training dataset, we averaged the p21-negative population to obtain ideal normal parameters and ranked p21-positive cells based on the intensity of staining, selecting the top 100 to define 'ideal' nuclear senescent features (Supplementary Fig. 14f). We used that information to generate a score based on a combination of nuclear features, able to evaluate senescence in individual cells ('cell senescence score', CSS, Supplementary Fig. 14g): if a nucleus has features like the model senescent population, the score assigned would be 1; if, on the contrary, the nuclear morphology is akin to a normal cell, the score assigned would be 0. Cells with more extreme senescence-like features would be assigned values >1.

We tested this classifier using a test set of p21$^{Cip1}$- stained liver sections (Supplementary Fig. 14h). The CSS performed well in predicting senescent cells in some fields but had low consistency and could not be used to predict senescence accurately at the cell level. To determine if the CSS could predict senescence at the tissue level, we analyzed the distribution of CSS scores in tissues with a relatively high (3.394%) or low (0.552%) percentage of p21$^{Cip1}$-positive cells. We analyzed the distribution of cells with high CSS values at different ranges (summarized in Supplementary Table 4). From that analysis, we conclude that calculating the percentage of cells with CSS between 1 and 5 correlated well with the percentage of senescent cells (as defined by p21-staining). This was exemplified by a higher fraction of cells with CSS 1–5 for a tissue with a higher percentage of senescent cells (Supplementary Fig. 14i–k). We subsequently adopted the percentage of cells in a tissue with a CSS 1–5 as a 'tissue senescence score' (TSS, as summarized in Fig. 7d).

To evaluate the tissue senescence score, we transduced constructs co-expressing GFP with oncogenic NRas (NRas$^{G12V}$, referred to as G12V) or an effector loop mutant (NRas$^{G12V, D38A}$, referred to as D38A) incapable of signaling downstream and induce senescence[35] (Fig. 7a). We performed immunohistochemistry (IHC) to detect GFP or p21$^{Cip1}$ in serial slides and stained another set of slides with hematoxylin (Supplementary Fig. 15a). Both cohorts had a similar frequency of GFP positive hepatocytes (Fig. 7e), but the frequency of p21$^{CIP1}$-positive senescent cells was higher in mice transduced with NRas$^{G12V}$ versus NRas$^{G12V, D38A}$ (Fig. 7f). Similar results were obtained when we used uPAR (Fig. 7g) or ORF-1 (Fig. 7h) as senescent markers.

We imaged hematoxylin-stained slides and calculated their CSS. Most cells had a higher CSS in NRas$^{G12V}$ liver sections compared to NRas$^{G12V, D38A}$ liver sections (Fig. 7i), corresponding to a higher TSS (Fig. 7j). We also calculated TSS scores from p21$^{CIP1}$-stained slides, which correlated with the frequency of p21$^{CIP1}$-positive cells (Supplementary Fig. 15b) and observed a significant correlation with the TSS calculated in hematoxylin-stained slides, suggesting that the classifier is consistent and antibody staining did not interfere with calculating senescence scores (Fig. 7k).

We further validate the approach using a different cohort of mice transduced with NRas$^{G12V}$ expressing constructs. We prepared serial liver sections, staining one with hematoxylin and co-staining the other with uPAR and p21$^{CIP1}$ to define senescent cells (Supplementary Fig. 15c). We confirmed a significant correlation between the TSS predictor and the percentage of senescent (uPAR-positive/p21$^{CIP1}$-positive) cells (Supplementary Fig. 15d).

## The tissue senescence score predicts senolytic drug efficacy in vivo

We previously showed that treatment with senolytic drugs such as ouabain reduced the percentage of senescent hepatocytes in mice that had undergone transposon-mediated transfer of NRAS$^{G12V}$ [38]. To investigate whether the TSS could reveal the effectiveness of senolytic compounds in vivo, we treated mice transduced with NRAS$^{G12V}$ with vehicle (DMSO) or a senolytic drug (Fig. 8a), sectioned the liver and performed IHC for p21$^{CIP1}$ and hematoxylin staining (Fig. 8b). As expected, the frequency of p21$^{CIP1}$-positive cells was lower in mice treated with the senolytic drug than in those treated with DMSO (Fig. 8c). We derived CSS from the hematoxylin-stained slides and observed lower scores for the liver sections from mice treated with senolytic drug than with DMSO (Supplementary Fig. 16a). The TSS was significantly lower in the senolytic-treated cohort, consistent with the reduced frequency of senescent cells (Fig. 8d). Again, we observed a significant correlation between the percentage of p21$^{CIP1}$-positive cells and TSS scores across the experiment (Supplementary Fig. 16b). Thus, the TSS correlates with other senescence markers and can be used to detect the efficacy of senolytic drugs in vivo.

## The tissue classifier predicts senescence-associated with liver fibrosis and aging in mice

Induction of senescence limits liver fibrosis[39] but lingering senescent cells accumulating in fibrotic sites can contribute to disease progression[40]. To determine if our senescence classifier could identify

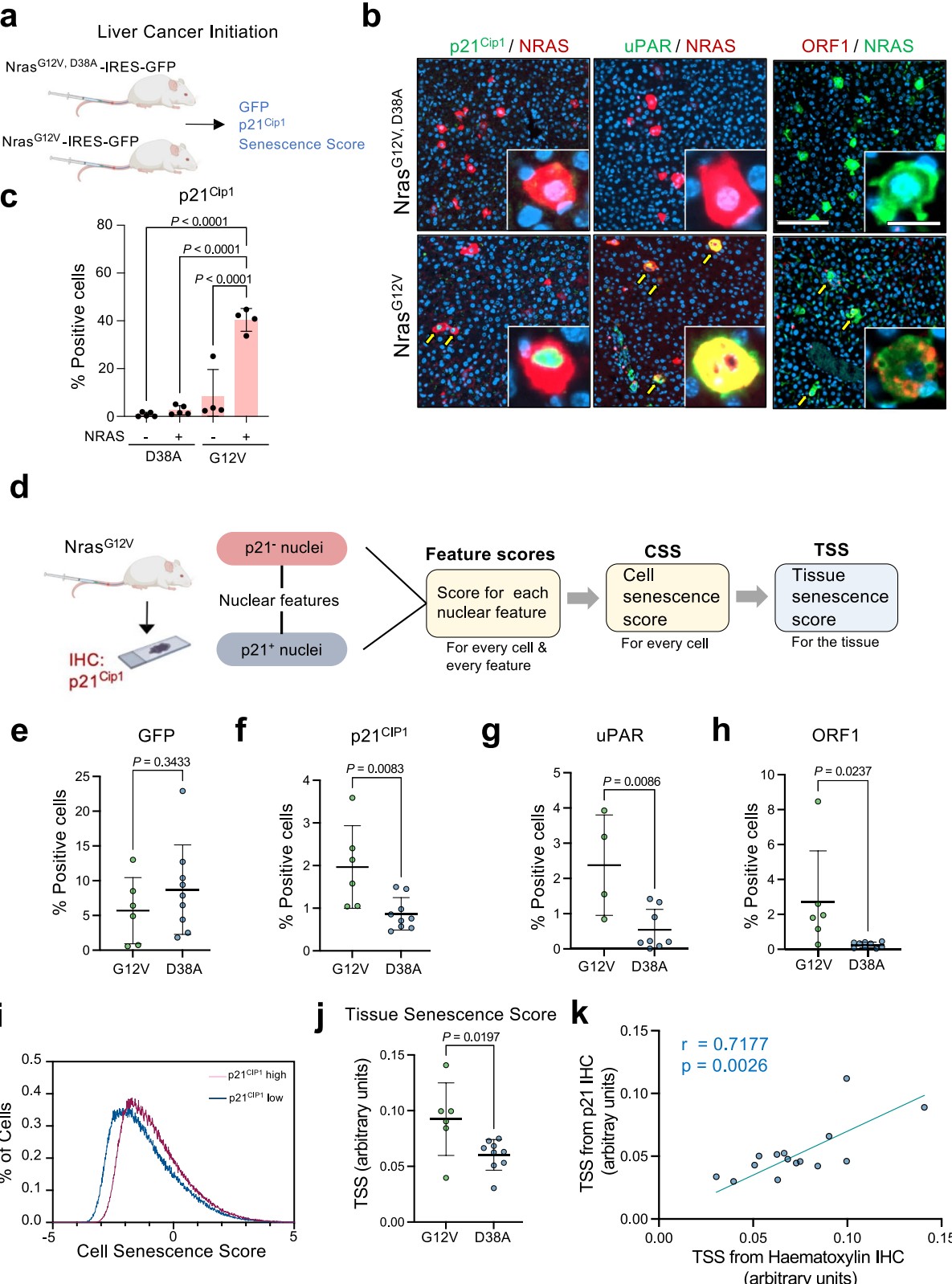

senescence-associated with fibrosis, we induced liver fibrosis in a cohort of mice by treating them with CCl₄ (Fig. 8e). As expected, IHC staining identified a significantly higher percentage of p21$^{Cip1}$-positive cells in the CCl₄-treated cohort when compared with oil-treated control mice (Fig. 8f, g). Importantly, the CSS distribution and TSS were also higher in the CCl₄-treated compared to oil-treated controls (Fig. 8h and Supplementary Fig. 16c).

To understand if our classifier could also distinguish age-related senescence, we took advantage of cohorts of young (~90 days old) and old (~600 days old) mice and performed p21$^{Cip1}$ immunostaining and hematoxylin staining of liver sections (Fig. 8i, j). We detected significantly higher percentages of p21$^{Cip1}$-positive cells in sections from old mice, consistent with an accumulation of senescent cells during aging (Fig. 8k). The CSS distribution and the TSS were also

**Fig. 7 | Tissue senescence score predicts senescence induced during liver cancer initiation. a** Experimental design for analysis of senescence in the liver cancer initiation model. **b** Representative images of liver sections obtained of mice transduced with Nras[G12V] or Nras[G12V, D38A]. Liver sections were co-stained with antibodies recognizing NRAS and p21[Cip1] (left), uPAR (middle), and ORF1 (right). Scale bar, 200 µm in the main picture and 20 µm in the insets. Yellow arrows mark double-positive cells. **c** Quantification of cells positive for p21[Cip1] as assessed by immunofluorescence in mice transduced with Nras[G12V]-ires-GFP (*n* = 4) or Nras[G12V, D38A]-ires-GFP (*n* = 5). GFP indicates NRAS-positive cells. Significance was calculated with a one-way ANOVA (Tukey's multiple comparisons test). Data represent mean ± s.d. **d** Experimental design for the development of a nuclear feature-based senescence scoring system in vivo. **e, f** Percentage of GFP (**e**) and p21[Cip1] (**f**) positive cells in the liver section of mice transduced with the Nras[G12V, D38A] (D38A, *n* = 9) or

Nras[G12V] (G12V, *n* = 6) expressing transposons. Significance was calculated using an unpaired two-tailed *t*-test. **g, h** Percentage of uPAR (**g**) and ORF-1 (**h**) positive cells in Nras[G12V]/Nras[G12V, D38A] mice. Significance for uPAR (Nras[G12V] *n* = 4; Nras[G12V,D38A] *n* = 8) and ORF-1 (Nras[G12V] *n* = 6; Nras[G12V,D38A] *n* = 10) was calculated using unpaired two-tailed *t*-test. **i** Distribution of cell senescence score in two sections corresponding to p21[Cip1] high (red) and low (blue) samples of the liver cancer initiation experiment. **j** Tissue senescence score calculated in hematoxylin-stained liver sections of mice transduced with the Nras[G12V, D38A] (D38A, *n* = 9) or Nras[G12V] (G12V, *n* = 6) expressing transposons. Statistical significance was calculated using unpaired *t*-test. **k** Pearson correlation coefficient (two-tailed, 95% CI) between tissue senescence score calculated in hematoxylin-stained and p21[Cip1]-stained liver sections (*n* = 15). *p* represents two-tailed nonparametric correlation probability. Data represent mean ± s.d; *n* represents number of mice. Source Data are provided in the Source Data File.

higher in the old versus young mice (Fig. 8l and Supplementary Fig. 16d). Altogether, these results indicate that our classifier can accurately predict senescence in different pathophysiological settings.

### The tissue classifier predicts levels of senescence in humans with non-alcoholic fatty liver disease

Senescence has been associated with fatty liver disease[39–41]. To investigate the applicability of our prediction models to patient samples, we analyzed liver sections resected from patients with mild non-alcoholic fatty liver disease (NAFLD, Fig. 9a). For each patient, we stained serial sections with either hematoxylin and eosin (H&E) or antibodies against p16[INK4a], as a surrogate of senescence (Fig. 9b). We calculated the CSS in H&E-stained samples and included for analysis those in which we were able to extract data from at least 10,000 cells (*n* = 34 patients). A comparison of the distribution of CSS from two samples with a relatively high and a relatively low percentage of p16[INK4a]-positive cells showed a similar shift in the high- p16[INK4a] sample as shown in the other models (Fig. 9c). Importantly, TSS showed a significant correlation with the percentage of p16[INK4]-positive senescent cells in those tissues (Fig. 9d). Overall, our results show that the TSS can be applied to predict senescence in human samples.

## Discussion

Senescent cells accumulate with age, are associated with multiple diseases, and are present in cancerous and fibrotic lesions[10]. Moreover, senescence is a novel therapeutic target with wide implications. Senolytic drugs have the potential to eliminate senescent cells involved in aging, cancer, and age-related diseases. However, a key factor limiting the quantification of senescent cells, and thus the identification and assessment of senolytics that induce their preferential killing, is the lack of universal and robust markers of senescence. We devised a family of machine-learning algorithms that take advantage of nuclear changes associated with senescence. These classifiers, rather than using images to feed neural networks, are based on a small number of interpretable nuclear parameters (that can be extracted with image analysis software), which standardizes and simplifies downstream analysis making it more amenable to be used by others.

Senescence is a heterogeneous response, and its characteristics might differ depending on the cell type and the stressor. Therefore, we developed classifiers trained with different cell types and stressors. Overall, our classifiers identified many types of senescent cells with good accuracy, even those types not included in their respective training sets. However, given the heterogeneity of senescence, our classifiers performed worse in identifying some types of senescence (e.g. in IMR90 cells or in response to barasertib treatment). In some instances, such as senescence induced by expression of constitutively activated MEK[42], the morphology of the nuclei is largely unchanged, so it is unlikely that our classifiers would work on those conditions, but

our study defines a framework that could be adapted to identify most types of senescent cells as needed. We compared several classifiers for their ability to identify a variety of senescent cells. Specific classifiers performed better in identifying certain types of senescence, but most of our classifiers were sufficient to predict senescence. Several of them, such as the GM senescence classifier, are consistent across the nine different datasets tested.

While ideally, one would want a 'perfect' predictor that can identify all senescent cells without false positives, such a toolbox might not exist due to the complexity in which senescence can develop. For a start, senescence is a heterogeneous and dynamic state, and the markers that are chosen to define senescence (in our case SA-β-galactosidase) will affect the comparisons. Importantly, imperfect predictions can be useful: we show that senescence-inducing drugs can be identified from predictions with not perfect accuracy but a high recall rate. As potential use cases for our algorithms, we show how they can characterize senolytic drugs (both in vivo and in vitro) and senescence inducers. To identify drugs that selectively induce senescence in tumors, we screened a collection of 676 drugs for their ability to induce senescence in a lung cancer cell line (A549) versus a normal lung fibroblast cell line (IMR90). We found a subset of drugs, including Eg5 inhibitors and multiple aurora kinase inhibitors, that preferentially induced senescence in A549 but not in IMR90 cells. Indeed, aurora kinase inhibitors were previously shown to induce senescence in cancer cells[32]. It will be of interest to use additional cell types in the future to elucidate the mechanisms underlying drug selectivity.

Another key use for senescence classifiers will be to predict senescence in tissue sections from preclinical mouse models and patient samples. We devised a tissue senescence score (TSS) that can identify cells undergoing oncogene-induced senescence during liver cancer initiation, and senescence in response to fibrosis and aging. Our TSS can also be used to assess the efficacy of senolytic drugs in vivo. Moreover, the TSS identified senescence in liver tissue sections of patients with mild fatty liver disease to an extent comparable to p16[INK4a] staining. While the TSS could be further improved and might need adaptation to identify senescence in other tissues, our results prove the utility of such classifiers to uncover pathophysiological senescence and assess potential senotherapies.

Neural networks were recently used to identify senescence from cell microscopy[18,19]. Given that the identification of senescent cells is a bottleneck in the field[43], our work described here joins those as a complementary and much-needed approach. Our approach starts from the pre-existing knowledge that senescent cells undergo chromatin rearrangement and changes in nuclear morphology[4,8,9] to devise machine-learning classifiers based on nuclear features. While starting from a different point, the reasoning behind our predictors converged with the observations of Heckenback et al.[19]. Moreover, open-source software such as CellProfiler[21] and Qupath[44] can be used to extract the

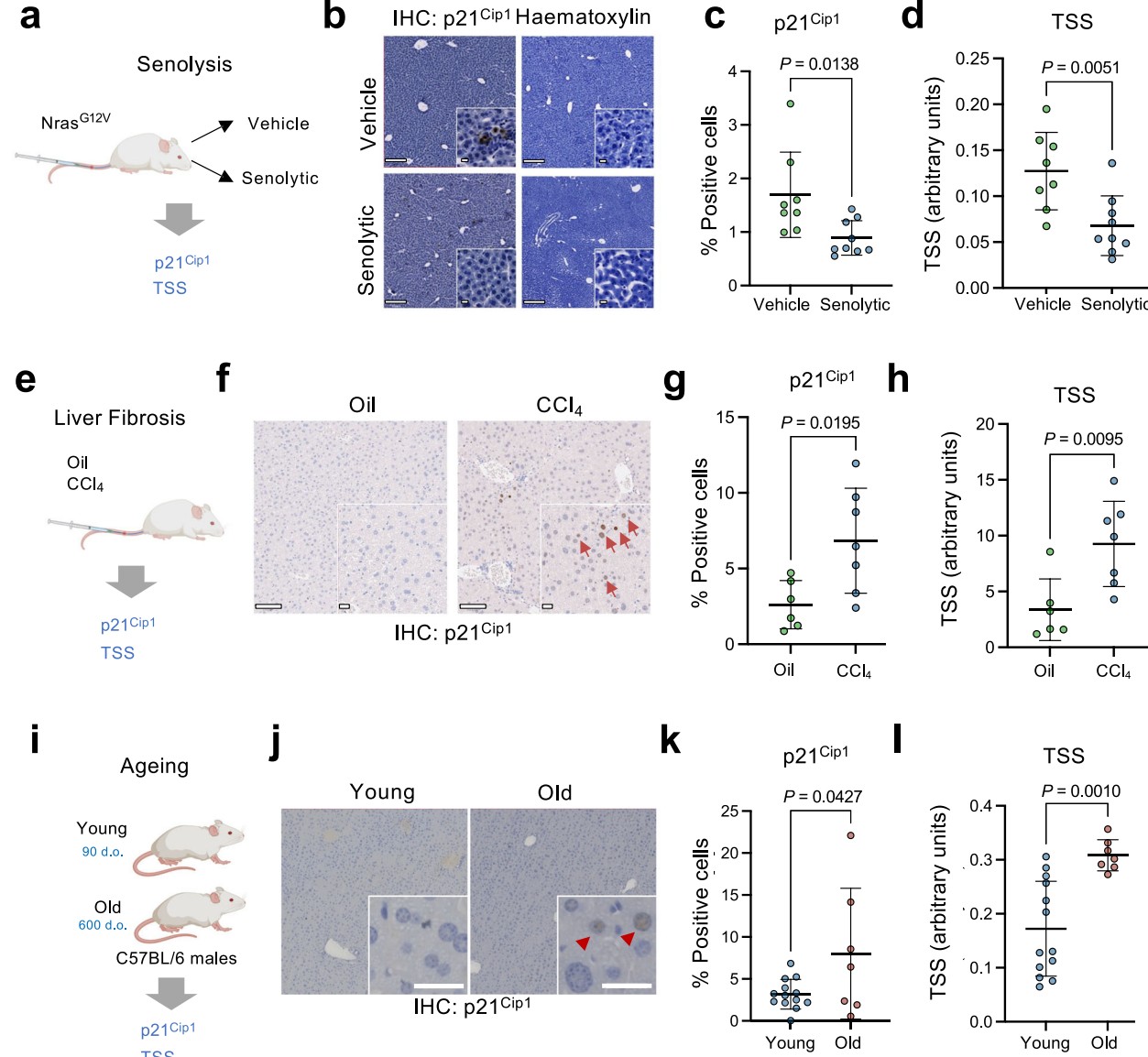

**Fig. 8 | Tissue senescence score predicts the effect of senolytic drugs and identifies senescence in liver fibrosis and aging. a** Experimental design for analyzing senescence in the senolysis experiment. **b** Representative images of p21$^{Cip1}$ and hematoxylin staining in liver sections. Scale bars are 100 μm in the main picture and 20 μm for the insets. The insets correspond to images from the same slides. **c** Percentage of p21$^{Cip1}$ positive cells in the liver section of mice transduced with a Nras$^{G12V}$ expressing transposon and treated with either vehicle ($n = 8$) or a senolytic drug ($n = 9$). **d** Tissue senescence score calculated in p21$^{Cip1}$-stained liver sections of mice transduced with a Nras$^{G12V}$ expressing transposon and treated with either vehicle ($n = 8$) or a senolytic drug ($n = 9$). **e** Experimental design for assessing senescence in the liver fibrosis model. **f** Representative images of p21$^{Cip1}$ IHC staining in liver sections. Red arrows mark p21$^{Cip1}$-positive cells. Scale bars are 100 μm in the main picture and 20 μm for the insets. **g** Percentage of p21$^{Cip1}$ positive cells in the liver section of mice treated with corn oil (oil) as a control ($n = 6$) or CCl$_4$ ($n = 7$) to induce liver fibrosis. **h** Tissue senescence score calculated in p21$^{Cip1}$-stained liver sections of mice treated with corn oil as a control ($n = 6$) or CCl$_4$ ($n = 7$) to induce liver fibrosis. **i** Experimental design for assessing senescence in liver sections during aging. **j** Representative images of p21$^{Cip1}$ stained in liver sections for young ($n = 13$) and old ($n = 7$) mice. Red arrows mark p21$^{Cip1}$- positive cells. Scale bar, 30 μm. **k** Percentage of p21$^{Cip1}$ positive cells in liver sections for young ($n = 13$) and old ($n = 7$) mice. **l** Tissue senescence score calculated in p21$^{Cip1}$-stained liver sections for young ($n = 13$) and old ($n = 7$) mice. Statistical significance for all comparisons was calculated using an unpaired two-tailed $t$-test. Data represent mean ± s.d; $n$ represents the number of mice. Source Data are provided in the Source Data File.

parameters needed for our classifiers, further facilitating its use by other labs.

In summary, we generated a family of algorithms that can predict senescence in multiple cell types and tissue sections by taking advantage of nuclear features. We provide proofs-of-concept demonstrating how these senescence classifiers can be used to identify and validate distinct senotherapies. Moreover, a tissue senescence score serves to evaluate senescence induction in tissue sections from preclinical mouse models and human patients.

## Methods

### Ethics

This research complied with all relevant ethical regulations and was approved and overseen by the following ethics review boards. Fully anonymized liver biopsies from patients with non-alcoholic fatty liver disease were obtained from the Imperial Hepatology and Gastroenterology Biobank which is fully REC-approved by the Oxford C Research Ethics Committee under REC reference 16/SC/0021. Informed written consent was provided by the donors. Mouse liver

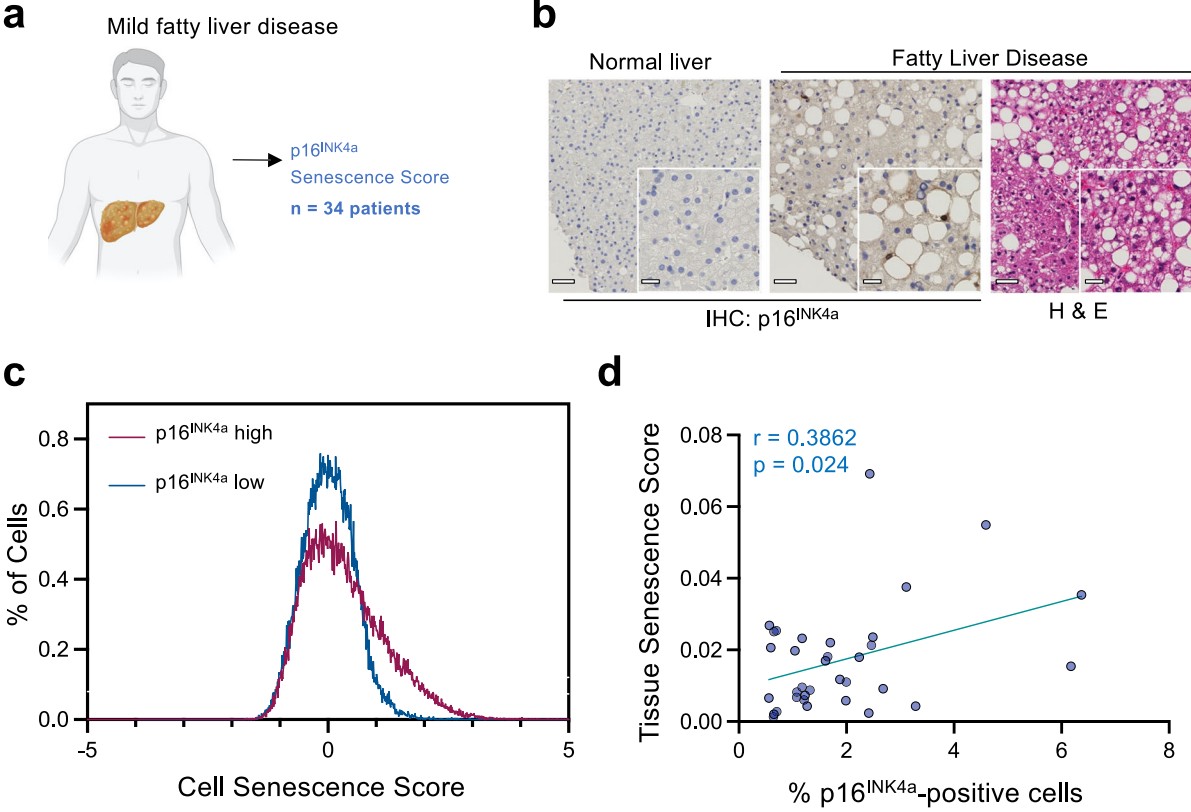

**Fig. 9 | Tissue senescence scores identify senescence in patients with non-alcoholic fatty liver disease (NAFLD). a** Schematic for the analysis of liver sections from patients with mild fatty liver disease. **b** Representative images of p16[INK4a] and H&E-stained liver sections in normal and fatty liver (out of 34) patients. Scale bar, 50 μm in the main image, 20 μm for the zoomed section. **c** Distribution of cell senescence score in two sections corresponding to samples with a high

(p16 = 9.0876%) and low (p16 = 0.2233%) percentage of p16[INK4a] positive cells. **d** Correlation between tissue senescence scores and percentage of p16[INK4a] positive cells ($n = 34$ patients). Pearson correlation coefficient (two-tailed, 95% CI) was used ($r = 0.3862$; $p = 0.024$). $p$ represents two-tailed nonparametric correlation probability. ($n = 34$ liver sections from different patients). Source Data are provided in the Source Data File.

fibrosis experiments were performed according to German law and with the approval of the Regierungspräsidium Karlsruhe (G139/19). All other mouse procedures were performed under license, according to the UK Home Office Animals (Scientific Procedures) Act 1986, ARRIVE 2.0, and approved by the Imperial College's animal welfare and ethical review body (aging experiments, PPL 70/8700; liver cancer initiation and senolysis experiments, PPL 70/09080).

**Cell lines**
Both female (IMR90, MCF7, HEK-293T) and male (A549, SK-HEP-1, HCT116, MRC-5) cell lines were used in this study. A549 (CCL-185), HCT116 (CCL-247), HEK-293T (CRL-11268), SK-HEP-1 (HTB-52), SK-MEL-103 (HTB-70), MRC-5 (CCL-171) and MCF7 (HTB-22) cells were obtained from the American Type Culture Collection (ATCC). Early passage IMR90 cells (ATCC CCL-186) were obtained from Coriell Institute. IMR90 ER:RAS cells were generated by retroviral infection of IMR90 cells and have been described elsewhere[45]. A549, HCT116, HEK-293T, IMR90, SK-HEP-1, SK-MEL-103, and MCF7 cells were cultured in Dulbecco's modified Eagle's medium (DMEM, Gibco), supplemented with 10% fetal bovine serum (FBS, Sigma F7524) and 1% antibiotic-antimycotic solution (Gibco). MRC-5 cells were cultured in Eagle's Minimum Essential Medium (EMEM), supplemented with 10% FBS and 1% antibiotic-antimycotic solution (Gibco). For inducing quiescence, the media was replaced with 0.5% FBS in DMEM.

**Senescence induction**
The following drugs were used to induce senescence in different cell lines after culturing cells in 96-well (Nunc, Thermo Fisher) or 100 mm

dishes (Corning, 430167): A549, 2 μM etoposide (Eto, Sigma–Aldrich, E1383), 0.2 μM doxorubicin (Doxo, Selleck chemicals, E2516), 2 μM alisertib (Ali, Selleck chemicals, S1133), 1 μM barasertib (Bara, Selleck chemicals, S1147); SK-MEL-103, 0.25 μM Eto, 0.1 μM Doxo, 0.25 μM Ali, 0.5 μM Bara; SK-HEP-1, 0.25 μM Eto, 0.1 μM Doxo, 0.1 μM Ali, 0.5 μM Bara; MCF7, 1.5 μM Eto, 0.1 μM Doxo. HCT116, 2 μM Eto, 0.5 μM Doxo. IMR90, 50 μM Eto, 2 μM Doxo (washed and media replaced 24 h post-treatment). Cell culture media with or without drugs was changed every 72 h. Senescence was assessed after 7 days of drug treatment unless otherwise stated.

**Screen for drugs inducing senescence**
676 drugs from the Target selective library (Selleck Chemicals) and Protein Kinase Inhibitor Library II (EMD Calbiochem®, Cat. No. 539745) were used to screen for drugs inducing senescence selectively in cancer cells. Drugs (at 10 μM) were added 24 h after seeding cells. Cell media was changed 72 h after and cells were fixed 5 days after cells were seeded. Cells were then fixed in 4% PFA, DAPI-stained, and images acquired (see High Throughput Microscopy). We screened the drugs in A549 lung adenocarcinoma and IMR90 fibroblasts in parallel, with biological triplicates per cell line. The percentage of senescent cells was calculated using the GM algorithm. A toxicity threshold was established against the viability of the positive controls. Samples with a lower cell count than 40% of that of the positive control were considered toxic and were excluded from analysis. This filtering excluded 69 drugs and data from cells treated with the remaining 607 drugs was taken forward for normalization. B-score normalization of the predictions of senescence induction was carried out (see B-score

normalization analysis). For a drug to be considered a hit, at least 2 of the 3 replicates would need to have a B-score >15. Drug candidates were then classified based on the predicted capacity to induce senescence in the cell lines tested based on the established threshold. Senescent B scores for all screened drugs have been included in the source data.

For follow-up studies (described in Fig. 6 and Supplementary Fig. 13), we used the following drug concentrations: doxorubicin, 0.5 μM; MLN8054 (Selleck Chemicals, S1100), 3 μM; ZM447439 (Selleck chemicals, S1103), 2 μM; SC144 (Selleck chemicals, S7124), 1 μM; ARQ 621 (Selleck chemicals, S7355); 0.1 μM; Niraparib tosylate (MK-4827, Selleck chemicals, S7625), 10 μM; AG-14361 (Selleck chemicals, S2178), 10 μM. Drug stocks (10 mM) were prepared in dimethylsulfoxide (DMSO) and stored at −20 °C.

## B-score normalization analysis
To analyze the drug screen, senescence prediction was normalized by B-score using the R package CellHTS2 (https://doi.org/10.18129/B9.bioc.cellHTS2)[46]. Value normalization was performed using the plate-averaging method and on separate batches for A549 and IMR90 cells.

## Antibodies
The following antibodies were used for the immunofluorescent and immunohistochemistry experiments: mouse monoclonal anti-bromodeoxyuridine (BrdU) (3D4; BD Biosciences, 555627) 1:2000; rabbit polyclonal anti-53BP1 antibody (Novus Biologicals, NB100-304) 1:1000; goat polyclonal anti-uPAR (Novus Biologicals, AF534), 1:200; mouse monoclonal anti-Nras (Santa Cruz, sc-31), 1:500; rabbit monoclonal anti-LINE-1 ORF1p (Abcam, ab216324) 1:500; mouse monoclonal anti-phospho-Histone H2A.X (Ser139) (Sigma−Aldrich, 05-636) 1:250; rabbit polyclonal anti-p21 (2947 S; Cell Signaling) 1:2000; mouse monoclonal anti-p53 (DO-1, Santa Cruz, sc-126) 1:100; rabbit monoclonal anti-p21 (EPR18021, Abcam) 1:700; rabbit recombinant monoclonal anti-GFP antibody [EPR14104] (ab183734) 1:500; Alexa Fluor 488/594 conjugated, (Thermo Fisher Scientific, A11029/A11032) goat anti-mouse IgG (H + L), 1:2000; Alexa Fluor 488/594 conjugated, (Thermo Fisher Scientific, A11034A11037) goat anti-rabbit IgG (H + L), 1:2000; Alexa Fluor 488/594 conjugated, (Thermo Fisher Scientific, A11055/A11058) donkey anti-goat IgG (H + L), 1:2000.

## Vectors
LentiGuide-Puro (Addgene, #52963) was used to express GFP, and pBabe puro IRES-mCherry (Addgene, #128038) for mCherry expression. Cells were FACS sorted and expanded. Cells expressing the construct were selected in 2 μg/mL of puromycin and kept under selection in 0.5 μg/mL of puromycin.

## Immunofluorescent staining of cells
Cells were grown in 96-well plates (Nunc™ MicroWell™, 167008, Thermo Fisher Scientific), fixed with 4% PFA (w/v) for 20 min, and then permeabilized in Triton X-100 0.2% diluted in PBS for 10 min. Cells were then blocked with 1% bovine serum albumin (BSA) (w/v) for 25 min. Cells were incubated with the primary antibody diluted in a blocking solution for 1 h and washed thrice in PBS. Cells were incubated with the secondary antibody (Invitrogen, Alexa Fluor™) diluted in blocking solution for 30 min and after washing thrice on PBS, 1 μg ml⁻¹ of DAPI was added for 12 min and washed with PBS thrice.

## Cytochemical SA-β-galactosidase assay
Cells were grown on 6-well (Nunc™, 140675) or 96-well plates and fixed with 0.5% glutaraldehyde (w/v) (Sigma−Aldrich) for 15 min, then washed with 1 mM $MgCl_2$/PBS at pH 6.0 and incubated in X-Gal solution (5 mM $K_3$(CN)$_6$, 5 mM $K_4$(CN)$_6$ and 1 mg ml⁻¹ of X-Gal by Thermo Scientific) for 6 h and 8 h in cancer cells and fibroblasts respectively, at 37 °C. Brightfield images were acquired using DP20

digital camera attached to an Olympus CKX41 inverted light microscope, at 4x and 10x magnification.

## Senolytic assay
Drugs were diluted to the required concentration in DMSO and stored at −20 °C. Cells were induced to senesce for 7 days and then cultured with 1 μM ABT-263 (Selleck Chemicals; S1001) or 5 μM ABT-737 (Selleck Chemicals; S1002) for 72 h. Cells were then fixed in 4%PFA (w/v) for 20 min and stained with DAPI for 12 min. Cells were then washed thrice in PBS.

## Irradiation-induced DNA damage
To induce DNA damage cells cultured in 96-well or 6-well plates were exposed to ionizing radiation (15 Gy). Media was changed 24 h later, and DNA damage was assessed by immunofluorescence at the required time point.

## Fluorescent SA-β-galactosidase assay
Cells grown in 96-well plates were washed in phosphate-buffered saline (PBS) and incubated with 33 μM $C_{12}$FDG (Abcam, ab273642) diluted in DMSO for 1 h. Cells were then fixed with 4% paraformaldehyde (PFA) for 15 min, washed thrice in PBS, and stained with 1 μg ml⁻¹ of DAPI. Images were taken using a high-throughput fluorescent microscope IN Cell Analyzer 2500HS (Cytiva) with a 10× objective for quantification. The percentage of SA-β-galactosidase positive cells was calculated using IN Carta™ Image Analysis Software (version 1.14) based on cellular fluorescence intensity using an arbitrary threshold to define positive cells.

## Gene expression analysis
Total RNA was extracted using RNeasy® Minikit (Qiagen). cDNA was produced using Superscript II reverse transcriptase (Invitrogen) and Random Hexamers (Invitrogen). Quantitative real-time PCR (RT-qPCR) was performed using SYBR™ Green PCR master mix (Applied Biosystems) in a CFX96 RT-PCR system C1000 Touch (Bio-Rad). For data normalization, GAPDH expression was used. The primer pairs used are:

*GAPDH*: GAAGGTGAAGGTCGGAGTC; TTGAGGTCAATGAAGGGG

*CDKN1A*: CGTGTCACTGTCTTGTACCCT; GCGTTTGGAGTGGTAG AAATCT

CDKN2A: CGGTCGGAGGCCGATCCAG; GCGCCGTGGAGCAGCAG CAGCT

*IL1A*: AGTGCTGCTGAAGGAGATGCCTGA; CCCCTGCCAAGCACAC CCAGTA

*IL1B*: TGCACGCTCCGGGACTCACA; CATGGAGAACACCACTTGTT GCTCC

## High-throughput microscopy
Cells were cultured in 96-Well Flat-Bottom plates (Thermo Fisher) and CellCarrier-96 Ultra Microplates (Perkin Elmer) were used. Plates were analyzed using IN Cell Analyzer 2500HS high content analysis (HCA) imaging at a magnification of 20× or 40×, with a binning of 1 × 1. TIF files obtained in HCA were analyzed using IN Carta™ Image Analysis Software (Cytiva, version 1.14). The acquired images had the following characteristics: width and length of 663.005 μm (2040 pixels), at a 3.0769 pixels per μm resolution (20×); width and length of 331.5 μm (1020 pixels), at a 0.3250024 pixels per μm resolution (40x). The following nuclear features were extracted using the In Carta software (See "Software" section): Area (in μm²), form factor (object roundness), elongation (object short axis/object long axis), compactness (average radius of the object), chord ratio (object min. chord ratio/object max. chord length), gyration radius (average radius of the shape), displacement (distance between the nucleus center of gravity and the cell center of gravity, normalized by the gyration radius of the nucleus). Where protein expression signal was analyzed intensity measures were also acquired.

To capture nuclear morphology parameter measurements, images were thresholded based on DAPI, primary objects identified, and measurements performed (see "Software" section). For quality control and to exclude artifacts, the cell segmentation pipeline performed a noise removal step, excluding shapes when image contrast was low. A sensitivity threshold was also established to accurately detect true nuclei events and a typical diameter of a nucleus was also established, to further exclude non-conforming structures. Overexposed targets were eliminated establishing a minimum DAPI intensity threshold and objects touching edges were excluded from the analysis to avoid partially acquired nuclei.

## Libraries of nuclear parameters of senescent and normal cells in culture

To develop the algorithm training sets (summarized in Supplementary Table 1), the indicated cells were seeded in 96-well plates and cultured and treated with the indicated drugs or DMSO as a control. Seven days after treatment, cells were fixed, and stained with DAPI for imaging. Each plate contained 30 wells with cells treated with the senescence inducer and 30 wells treated with DMSO (as a control). Datasets for each cell line and senescence induction contained data derived from at least three plates, resulting in a total of between $0.1 \times 10^6$ and $0.9 \times 10^6$ cells per condition (see Supplementary Table 1 for details). Training datasets (as indicated in Supplementary Table 1) were then generated by randomly selecting 10,000 normal cells and 10,000 treated cells for each training set. For the General Model, randomized samples from all training datasets (3 cell lines, A549, SK-MEL-103, SK-HEP-1; 4 conditions: Etoposide, Doxorubicin, Alisertib, Barasertib) were taken to develop the classifier, as noted in Supplementary Table 1. Independent training sets (with different randomizations of the same libraries) were constructed for the classification tree and random forest algorithms.

## Software

The following packages were utilized for classification trees, random forest building, and related analysis. For Classification tree (CT) and random forest (RF) algorithms python version 3.7.7 was used. The following packages were also utilized: scikit-learn and derived packages;[47], pandas, numpy, matplotlib. pyplot, seaborn, and csv. Area, Form Factor, Elongation, Compactness, Chord Ratio, Gyration, and Displacement were used as nuclear features. Analysis of public software CellProfiler (version 4.2.4)[21] was performed using a nucleus detection workflow. For High Content Analysis (HCA), InCarta software (Molecular Devices, version 1.14) was used. For B-score analysis R (version 4.3.1) and packages BiocManager (version 1.30.22) and cellHTS2 (version 2.64) were used.

## Generation of classifiers to identify senescence in cell culture

For classification tree (CT)-based classifiers, preliminary classification trees were built using sklearn, providing 30% of the training set as test size. After assessing initial accuracy, AUC, and ROC curves, cost complexity pruning was performed to avoid overfitting. The optimal value of alpha was calculated and applied to develop the classification trees. Obtained classification tree branches were eliminated where redundancy occurred. For random forest (RF)-based classifiers, the test size was set at 0.5. For CT classifiers the classes were established in a binary manner, where 0 equalled growing, normal cells and 1 represented senescent (treated) cells. For RF classifiers senescence probability was estimated and values > 0.5 were considered as senescent. Classifiers were ultimately tested on new experiments (test data) and the accuracy of prediction was assessed. For the Voting-Based Clustering Algorithm (VCA), the input from all algorithms described in Supplementary Table 2 (except the CellProfiler-based classifiers) was considered. Relabeling of partitions was avoided and opted for a democratic vote system with equal weight per classifier algorithm.

## Algorithm performance metrics

Algorithm accuracy on the test data was measured by area under the curve (AUC) and receiver operating characteristic (ROC) curve (True Positive Rate vs False Positive Rate) assessment and posterior testing on new data not belonging to the training datasets. Algorithm accuracy was also measured using the following metrics in a dataset of co-cultures of senescence and normal cells:

$$Accuracy = \frac{TN + TP}{TN + TP + FP + FN} \tag{1}$$

$$Precision = \frac{TP}{TP + FP} \tag{2}$$

$$Recall = \frac{TP}{TP + FN} \tag{3}$$

$$F1\ Score = \frac{Precision \times 2Recall}{Precision + Recall} \tag{4}$$

TP true positive; TN true negative; FP false positive; FN false negative.

## Co-cultures of senescent and non-senescent cells

The setup of these experiments is described in detail in Supplementary Fig. 5. Briefly, $10^5$ cells (for DMSO) and $10^6$ cells (for treatment) were seeded in 100 mm dishes. 24 h after plate seeding media was washed once with PBS (Gibco™, Thermo Fisher, 10010023), and DMSO or senescence-inducing drug was added to the media (DMEM, 10% FBS, 1% antibiotic-antimycotic). Media (with drug or DMSO) was replaced every 72 h. 6 days post-treatment, senescent and control cells were trypsinized and counted using a Guava Muse Cell Analyzer. DMSO-treated and drug-treated cells were seeded in separate master plates (96-well Round (U) Bottom Plate, Thermo Fisher, 163320) as indicated in Supplementary Fig. 5c. Those master plates were used to generate the co-culture plate. After 24 more hours, cells were cultured in $C_{12}FDG$ (33 μM, diluted in DMSO) for 30 min, fixed in 4% PFA for 15 min, and stained with DAPI.

## Senescence classifiers based on CellProfiler data

To develop algorithms utilizing CellProfiler (version 4.2.4)[21], we produced a new training set consisting of 4 plates of DMSO-treated and etoposide-treated A549 cells. 7 days after treatment cells were fixed, stained with DAPI, and imaged using an INCell Analyzer 2500HS. The acquired TIF files were then analyzed utilizing a bespoke nuclear workflow CellProfiler protocol (Dapi_CellProfiler.cpproj, see "Code Availability" section). The following features were considered for CT and RF algorithm development: Area, Bounding Box Area, Compactness, Convex Area, Eccentricity, Equivalent Diameter, Extent, Form Factor, Major Axis Length, Maximum Feret Diameter, Maximum Radius, Mean Radius, Median Radius, Minimum Feret Diameter, Minor Axis Length, Perimeter, and Solidity. To capture nuclear morphology parameter measurements, images were thresholded based on DAPI, primary objects identified, and measurements performed. From the resulting parameter files, cells were grouped by treatment, and randomized, and 10,000 cells were extracted per condition, following the same procedure to develop CT- (AECP) and RF-based (AERFCP) classifiers.

## Mouse experiments

Mice were kept on a 12-h light/dark cycle and between 21–23 °C temperature and 45–65% humidity levels under specific pathogen-free barrier conditions within individually ventilated cages with *ad libitum* access to standard chow food (SDS RM1/3 [E] LBS Serving Biotechnology) and water. C57BL/6 J littermate mice were used unless

otherwise specified. As sex is not a factor in the scope of the study design, liver sections of both male and female mice were used (as detailed below) for different experiments. Animal welfare was monitored and euthanasia practices were performed according to the requirements of the aforementioned practice licenses and regulatory frameworks.

## Liver fibrosis

Eight-week-old male C57BL6/J mice were treated twice a week with either corn oil ($n = 6$) or carbon tetrachloride CCl$_4$ (0.5 mL/kg) ($n = 7$) by intraperitoneal injection for 6 weeks to induce liver fibrosis as described before[48]. Mice were sacrificed at the indicated time points and analyzed for senescence.

## Aging experiment

Male C57BL/6 J littermates were used. Mice that were 90 days old ($n = 13$) were used for the young cohort and 600 days old mice ($n = 7$) for the old cohort. Mice were sacrificed at the indicated time points and analyzed for senescence.

## Liver cancer initiation and senolysis experiments

Hydrodynamic tail vein injection (HDTVI) was carried out in female C57BL/6 J (Charles River UK) mice aged 5–6 weeks using 25 µg of a transposon expressing Nras$^{G12V}$ or Nras$^{G12V, D38A}$ along with 5 µg of SB13 transposase-expressing plasmid. All plasmids were prepared with GenElute HP Endotoxin-Free Maxiprep kit (Sigma). For HDTVI, vectors were diluted in normal saline to a final volume of 10% body weight. HDTVI was performed within 7–8 s.

For liver cancer initiation experiments, mice transduced with transposon vectors co-expressing GFP and either Nras$^{G12V}$ ($n = 6$) or Nras$^{G12V, D38A}$ ($n = 9$) were used.

For the experiment described in Supplementary Fig 15c, d an additional cohort of mice transduced with transposon vectors co-expressing GFP and Nras$^{G12V}$ ($n = 12$) was used, Mice were culled 9 days (after HDTVI) and livers were collected for paraffin embedding.

For senolysis experiments, a transposon vector co-expressing Nras$^{G12V}$ and Gaussia luciferase (Gluc) was used. On day 5 after HDTVI mice were given either a senolytic drug ($n = 9$) or vehicle ($n = 8$) intraperitoneally (i.p.) daily for 4 days. 24 h after the last drug injection mice were culled and livers collected for paraffin embedding.

## Immunohistochemical staining of tissue sections

Mouse liver tissue sections were deparaffinized in Histoclear™ (Scientific laboratory supplies) for 5 min, and washed in decreasing concentrations of ethanol, until a final 5 min wash in dH$_2$O Heat-induced epitope retrieval (HIER) was then performed in a pressure cooker for 20 min in citrate-based at pH 6.0 (VectorLab, H-3300-250) or tris-based at pH 9.0 (VectorLab, H-3301-250), following the antibody manufacturer's instructions. For intracellular expression stains and sections were washed in Triton X-100 0.2% in PBS for 10 min and washed in PBS for 5 min. Slides were then incubated in BLOXALL blocking solution (VectorLab, SP-6000), washed in PBS, and exposed to Animal-Serum Free serum (Cell Signaling, 15019 L) diluted in dH$_2$O for 30 min. Slides were then incubated with primary antibody overnight in a humidified chamber at 4 °C. Slides were washed twice in PBS for 5 min and incubated with secondary antibody SignalStain® Boost IHC detection reagent Mouse/Rabbit, HRP (Cell Signalling Technology, 8125) for 30 min. After, slides were washed in PBS and incubated in SignalStain DAB (CST, 8059) for 5 min or until the HRP signal was visible and the reaction stopped in dH$_2$O. Cells were then stained for Hematoxylin (DAKO, Mayer's Hematoxylin, S3309) for 30 s and washed in dH$_2$O. When necessary, slides were further stained in Eosin Y (Sigma−Aldrich, HT110132-1L). Slides were, dehydrated in 75% ethanol for 1 min and 100% ethanol for 5 min, washed in Histoclear for 5 min, and mounted in DPX (Sigma−Aldrich).

## Immunofluorescence staining of tissue sections

For Immunofluorescence staining, deparaffinization, and antigen retrieval were performed as described previously (see "Immunohistochemical staining of tissue" sections). Mouse liver samples were incubated overnight in the primary antibody previously diluted in antibody diluent (Dako). Samples were washed in PBS three times for 5 min. Samples were then incubated in secondary antibody Signal-Stain® Boost detection reagent (Cell Signalling Technology, 8125) for 45 min. The signal was then amplified using Thermo Fisher AlexaFluor™ 488/647 Tyramide SuperBoost™ Kit (B40958) following the manufacturer's instructions. To perform double staining, samples were incubated in HCl 0.02 N for 20 min after the first antibody signal amplification step. Samples were then washed in PBS for 5 min and peroxidase blocking was reapplied for 20 min. Samples were then incubated in Animal-Serum Free blocking solution (Cell Signaling, 15019 L) diluted in H$_2$0 for 1 h and the second primary antibody incubation was performed overnight. The signal was then amplified using a different wavelength-reactive SuperBoost™ Kit, using antibodies raised in different hosts to avoid cross-reactivity. Samples were then washed three times in PBS for 5 min and incubated with DAPI (1 µg ml$^{-1}$ in PBS) for 5 min. Samples were washed thrice in PBS for 5 min and mounted in 50% glycerol in PBS.

## Slide image acquisition and analysis

Slides containing preclinical and clinical liver samples were acquired using 40x brightfield objective on a Zeiss AxioScan Z.1 or Leica Aperio AT2 slide scanner and analysis was performed using QuPath version 0.3.0, adjusting the built-in cell acquisition parameters to immunofluorescent and immunohistochemical samples to maximize the accuracy of cell and nuclear detection for feature extraction and signal quantification. A pixel size of 0.5 µm was established, and to accurately detect nuclei and avoid artifacts a background and median filter radius was established, together with a Gaussian filter to reduce noise and a minimum nuclear area. To further ensure accurate detection, an intensity threshold and a maximum background intensity were set. For immunofluorescence nuclear detection, DAPI was used as a detection channel and the same artifact filters were incorporated into the pipeline. The following features were extracted: centroids X and Y (coordinates in µm, for single cell positioning in the slide), nucleus area, nucleus perimeter, nucleus circularity, nucleus max caliper, nucleus min caliper, and nucleus eccentricity. Where immunohistochemical staining was performed, DAB optical density (OD) mean and total DAB were also acquired (nuclear of cellular, corresponding to protein expression localization). The indicated data were then extracted and analyzed. For human patient samples, a circularity threshold (nuclei superior to 0.7) was established to ensure that captured nuclei belonged predominantly to hepatocytes and not to fibroblasts or immune cells. The number of cells per sample (as evaluated by acquired nuclei) varied between $7 \times 10^4$ and $1 \times 10^5$. Samples with less than 10,000 cells were excluded from the analysis.

## Senescence scoring system in tissue sections

To assess senescence in liver tissue sections, we took advantage of liver sections of mice transduced with Nras$^{G12V}$ using hydrodynamic tail vein injection (HDTVI), stained with anti-p21$^{Cip1}$ antibodies, and counterstained with Hematoxylin. 4 slides were scanned and analyzed using QuPath. Cells were classified as p21$^{Cip1}$-positive or -negative based on DAB nuclear mean intensity (>0.2 for p21$^{Cip1}$-positive cells and <0.2 for p21$^{Cip1}$-negative cells). Nuclear features (area, perimeter, maximum caliper, minimum caliper, eccentricity, and circularity) were extracted for p21$^{Cip1}$-positive and p21$^{Cip1}$-negative cells (see "Slide image acquisition and analysis" section). Data from $4.32 \times 10^5$ p21$^{Cip1}$-negative cells was used to obtain average measurements, which defined the ideal normal (non-senescent) cell. p21$^{Cip1}$-positive cells were ranked based on the p21$^{CIP1}$ staining intensity. The average parameters of the top one

hundred p21$^{Cip1}$-positive cells were selected to define the ideal senescent cell. Therefore, for each nuclear feature, an ideal parameter value for normal cells (P$^N$) and an ideal value for senescent cells (P$^S$) were defined (shown in Supplementary Table 5).

For each cell, we performed the following operation:

$$\frac{\sum_{i=0}^{n}\left(\frac{p-P^N}{(P^S-P^N)}\cdot\frac{\left|\frac{P^N}{P^S}\right|}{\sum T}\cdot\frac{1}{n}\right)_n}{n} \tag{5}$$

Where:

$n$ = number or features
$p$ = acquired parameter value
$P^N$ = ideal parameter value for normal cells
$P^S$ = ideal parameter value for senescent cells
$T$ = summation of absolute values of differences from acquired parameters

Consequently, individual features from single nuclei obtained a feature score, which was then corrected by the weighted effect of each feature in the summation of the absolute values of the differences (ΣT), which allows to proportionalise the effect of the features. Individual feature scores were then aggregated to provide a single value (that we termed the cell senescence score, CSS) for each cell. To calculate the tissue senescence score (TSS), for each tissue section, we plotted curves showing the distribution of individual cell senescence scores for all the cells present in that section. Thus, the TSS from a given tissue sample relies upon the distribution of its individual CSS values. To obtain the TSS value, that describes senescence presence in the sample, we scored the sections based on the percentage of CSS values between 1 and 5 (CSS values associated with nuclear senescent features), thus obtaining a unique metric for the tissue section. We evaluated other ranges of CSS values (as shown in Supplementary Table 4) and chose 1–5 as the one better correlating with the percentage of senescent cells present in the tissues. Importantly, the same operation and metrics to calculate CSS/TSS values (without adjusting or changing any of the parameters for subsequent experiments) were applied to all preclinical and clinical liver models used in this study.

**Senescence assessment in samples from patients with mild fatty liver disease**
Human liver biopsies were fully anonymized and acquired from the Imperial Hepatology and Gastroenterology Biobank, therefore no regard for sex and gender was considered. Sections were deparaffinized, and hydrated, and then heat-mediated antigen retrieval was performed in citrate-based pH 6.0 solution. The endogenous peroxidase was quenched with 3% hydrogen peroxide. The sections were incubated with mouse monoclonal to p16$^{INK4a}$ (CINtec, 9511, clone E6H4), followed by rabbit anti-mouse IgG. The sections were subsequently incubated with anti-rabbit IgG conjugated with polymeric horseradish peroxidase linker (Leica Bond Polymer Refine Detection, DS9800). DAB was used as the chromogen and the sections were then counterstained with hematoxylin and mounted with DPX. IHC was performed on Leica BOND III. Serial sections were stained with H&E and used to calculate tissue senescence scores. Slides were scanned with NanoZoomer 2.0HT (Hamamatsu, Japan). NDP.scan 3.2.12 software was used for digital image acquisition and NDP.view2 software was used for image viewing. 36 samples were processed and imaged, but 2 samples with less than 10,000 cells were excluded from the analysis.

**Statistical analysis**
We used GraphPad Prism (Version 9.4.0) for statistical analysis. Two-tailed, unpaired Student's $t$-tests were used to estimate statistically significant differences between groups, as well as one-way ANOVA when required. Pearson correlation analysis was performed utilizing a two-tailed option, with a 95% confidence interval. Simple linear regression was also performed to display the corresponding fit line. To study the cumulative distributions between treated and control nuclear features we performed the Kolmogorov-Smirnov (K-S) test and detailed the maximum absolute difference (D) and the associated $P$ value.

**Reporting summary**
Further information on research design is available in the Nature Portfolio Reporting Summary linked to this article.

## Data availability
Source data are provided with this paper.

## Code availability
Custom code and training sets can be found at: https://github.com/Sen-Lab-LMS/Senescence_nuclear_features [49], which is archived in Zenodo with the identifier [https://doi.org/10.5281/zenodo.10499895].

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

## Acknowledgements

We are grateful to members of J. Gil's laboratory for reagents, comments, and other contributions to this project. We also thank Andre Brown (MRC LMS), Peter Sarkies (Oxford University), Mikhail Spivakov (MRC LMS), Aldo Faisal (Imperial College) and Anil Bharath (Imperial College) for advice and discussion, and Angela Andersen (Life Science Editors) for editing the manuscript. Figures 1a, 1g, 2a, 2f, 2i, 3a, 3f, 3h, 4a, 5a, 5f, 6e, 7a, 7d, 8a, 8e, 8i, 9a, and Supplementary Figs. 2b, 5a–c, 6e, g, 7a, 8a-b, 9a–d, 10a, b, 12a and 14c have been partially generated with Biorender (Biorender.com). For open access, the author has applied a Creative Commons Attribution (CC BY) license. Core support from MRC (MC_U120085810) and CRUK (C15075/A28647) funded this research in J. Gil's laboratory. M. Heikenwalder was supported by a European Research Council (ERC) Consolidator grant (HepatoMetaboPath), SFB/TR 209 project ID 314905040, SFB 1479 (Project ID: 441891347), the Rainer Hoenig Stiftung, Research Foundation Flanders (FWO) under grant 30826052 (EOS Convention MODEL-IDI) and a seed funding from HI-TRON. Work was supported by the Medical Research Council grant MC-A654-5QB40 and a Wellcome Trust Grant 098565/Z/12/Z in D. J. Withers' lab. S. Vernia was supported by MRC funding (grant number MC_UP_1102/18). H. Kudo is supported by the NIHR Imperial Biomedical Research Centre (BRC).

## Author contributions

I.D. designed, performed, and analyzed the cell culture experiments, stained and analyzed the cancer initiation, senolysis, and aging experiments, developed the machine-learning algorithms, and helped write the first draft of the manuscript. J.P. performed, processed, and stained cancer initiation, senolysis, and aging experiments. B.S. conceived and performed liver senolysis and cancer initiation experiments. S.G. conceived and performed aging and fibrosis experiments. H.K. stained and imaged patient samples. D.H.M. conceived and performed liver senolysis and cancer initiation experiments. L.B. conceived and performed immunofluorescent experiments in mice. J.E.B.A. performed immunohistochemical analysis and assisted and imaged the fibrosis experiment. S.V. conceived, and secured funding and approval for the senolysis and cancer initiation experiments. D.J.W. conceived, and secured funding and approval for the aging experiments. R.D.G secured funding for fatty liver disease experiments and was responsible for the selection and histological analysis of the human liver biopsies. P.M. and R.F. selected and collected human liver biopsies. M.H. conceived, and secured funding and approval for the fibrosis experiment. J.G. conceived and

designed the project, secured funding, and wrote the manuscript, with all authors providing feedback.

## Competing interests

J.G. has acted as a consultant for Unity Biotechnology, Geras Bio, Myricx Pharma, and Merck KGaA. Pfizer and Unity Biotechnology have funded research in J.G.'s lab (unrelated to the work presented here). J.G. owns equity in Geras Bio. J.G. is a named inventor in MRC and Imperial College patents, both related to senolytic therapies (the patents are not related to the work presented here). The remaining authors declare no competing interests.
