## [Peer Review File · Nature Communications]

REVIEWER COMMENTS

Reviewer #1 (Remarks to the Author):

The authors have addressed my concerns. This is an interesting and potentially broadly useful study.

Reviewer #2 (Remarks to the Author):

NCOMMS-23-29145-T

After reading the revised manuscript and the responses to my comments, I've noticed that the authors have effectively addressed the majority of my concerns. As a result, I firmly believe that the manuscript has significantly improved and is deserving of publication in Nature Communications.

However, there is one last point I would like to point out. It is widely known that, unlike other methods of inducing cellular senescence, when cellular senescence is induced by constitutively active MEK, cellular and nuclear morphology remain largely unchanged (DOI: 10.1101/gad.12.19.3008). Hence, it would be interesting to know if the classifier developed by the authors in this study can be applied to determine senescent cells in such cases. Therefore, it would be better for the authors to mention something on this point in the Discussion section.

Reviewer #3 (Remarks to the Author):

Manuscript review

Title: Detection of senescence using machine learning algorithms based on nuclear features.

Summary

I have reviewed the work of Duran et al. with specific focus the data science, computer vision and result visualisation aspects presented in the paper.

The work can be divided in to two parts. In part 1, the authors develop a set of classifiers for the identification and assignment of tissue culture cells to a phenotype of choice (i.e. senescent, non-senescent). In part two, the authors develop a Cell and Tissue Senescence Score (CSS and TSS respectively), which they use to describe senescence in tissues

The focus of the paper is on the methodology rather than the description of a biological discovery. Unfortunately, many key aspects of Methods-type manuscript are underdeveloped here and would need to be included to justify publication in this journal. Further, I think that there are some technical mistakes which needs addressing to ensure that the proposed method is truly adding value to the research community.

I applaud the authors for showcasing models for different types of primary data such as tissue culture cells and tissue sections and multiple application settings. However, I can't shake the feeling that this breadth of examples comes at cost of quality and thoroughness. All parts of the manuscript have multiple, severe data science issues which raise quality concerns.

In its current form, I would not recommend the publication of this work. The authors would have to make very substantial changes to the overall structure and content of the manuscript to convince me that the data science in this work meets the standard Nature Communications readers have grown accustomed to.

On the structure of the manuscript

The manuscript is too long. I would encourage the authors to consolidate their findings in a total of 4 figures. This would help focus the message of the paper and limit redundancy and the potential for confusion. Specifically, I would suggest that the authors:

1. Try to emulate the structure of a methods-focused manuscript better. E.g. Introduce the new methods, show its methodological soundness, benchmark it against existing solutions, show its utility in the discovery of biological results. Many excellent example can be found in Nature Methods.
2. Pick one classifier to highlight in the main manuscript. I appreciate the effort of trying multiple classifiers but a) it isn't surprising that multiple work b) it isn't clear to me what the value is of showing multiple classifiers ins such depths, give that the authors only report results achieved with them rather than elaborating on the differences between the results generated by the classifiers. In essence, the authors call different functions of the sklearn python package. This isn't particularly innovative nor difficult.
3. For this flavour of publication, the methodology is overall poorly described in the main manuscript and particularly in the Methods section (what to describe is suggested further down in this document).

On the comparison of ML algorithms with human curated features vs. Neuronal Networks

The authors discuss multiple times the pros and cons of classifiers versus Neuronal Networks. To my understanding, this comparison is particularly motivated by a recent paper using Neuronal Networks to identify senescence in tissue culture cells and senescence. I would advise the authors to stay clear of a comparison which isn't grounded in quantifications since either side has its benefits and shortcomings. I would expect a more balanced and critical discussion of their methodology, should the authors insist on including a pure qualitative comparison in their manuscript.

On the classifiers presented in part 1

In the first part of the manuscript, the authors develop a suite of classifiers which use nuclear morphology features to assign tissue culture cells to two classes (senescent, non-senescent). The classifiers are used on data generated and extracted from multiple microscopes and image analysis software packages. In Figures 1, 2, and 3a) to 3i) the authors showcase a set of classifiers trained on perturbation data that results in mixed populations of senescent and non-senescent cells thus introducing class errors in the ground-truth data.

1. The quality of the classifier immediately drops substantially when the authors apply the model in a test scenario. E.g. Figure 1h and 1i detecting ~90% of cells as senescent in train (AEM) and only 65% in test (AEM). While the authors are right that there is still a significant difference between DMSO and Eto treated populations in train and test, their performance drops by almost 30% ($1 - (65\% / 90\%)$). A similar performance drop can be seen in Figure 2 d) and e). This is normally a sign of a poor classifier. The authors don't report this performance drop properly in the result section nor do they touch on it in the discussion.

2. In Figures 1, and 2, the authors repeatedly compare the classifier results to the % of SA-B-Gal positive cells. It would improve the legibility of the figures and facilitate the comparison immensely if these results would be displayed on the same Y axis.

They recognise the limitation of these classifiers and implement a fluorescence marker based class assignment of cells as ground-truth. They then compare these improved classifiers to the one of the best performing original classifiers and argue that given the similarity in Precision, Recall, Accuracy and F1 scores, the original and improved classifiers are of the same quality (I am paraphrasing here). The authors then proceed to use the original classifiers for the analysis of a drug screen. The comparison between original and fluorescence-guided classifiers is too simplistic and potentially misleading.

I would ask the authors to address these questions and requests:

3. The presented metrics should only be compared between classifiers if the classifiers have been evaluated on a shared set of cells. Whether this was done, is not explained clearly. NB, that if the metrics come from different sets of cells the comparison is meaningless. Please clarify.

4. In fact there is only one valid comparison setup to justify their conclusion of using the original classifiers interchangeably with the fluorescence-guided ones. Namely to use the fluorescence-assisted assignment of senescence should serve as ground truth. Thus, I would kindly ask the authors to define a set of cells from said ground truth and apply the "AEM" and the new classifier and evaluate the results

using the previously mentioned metrics. This analysis good also be repeated multiple times to statistically test whether “AEM” is indeed as good as the “newer” classifiers. Pending the proposed analysis, no comparison should be drawn between the classifiers.

5. The authors do not describe or show any quality control efforts of the primary and derived data. Given the focus of the work this is surprising. I would kindly ask the authors to include a QC pipeline for all their cell experiments, which at the minimum showcases their ability to exclude imaging artefacts in the images, their ability to segment nuclei accurately, and their ability to exclude miss segmented nuclei. Providing the readers with such information is crucial to raise the confidence in the presented data and algorithms.

6. Many of the experiments are control experiments and should be presented in the Extended Data or Supplementary section of the work.

7. The authors focus solely on morphology features. They could have included nuclear DAPI intensity and texture features (they are ready available in a CP pipeline). Could they elaborate why they chose not to do this and or show that the inclusion of such features doesn't improve the classifier performance.

The authors then apply multiple classifiers on various pharmacologically perturbed and unperturbed cell lines (Figures 4 and 5).

8. These are a great result, as the authors show the ability of the classifier to be used on unseen data, i.e. different cell lines.

9. However, it also shows the clear limits of a pretrained classifier (Supplementary Figure 5b and 6). The authors have trained a classifier on A549 cells treated with Etoposide (AEM). When applied to A549 treated with another senescence inducing drug (Barasertib) or on another cell line the performance drops substantially. To claim that “The above results suggest that senescence classifiers are generalizable to different stresses and cell types, beyond those used to train the algorithms.” Isn't well substantiated as there are striking indications that AEM, in fact, does not generalise.

10. I do not understand the purpose and methodology of Figure 5. Why so many classifiers and how where there trained. I would kindly aske the authors to explain this experiment better. The authors use the GM model later in the manuscript. Maybe these tables could go supplementary.

On drug discovery/ drug screen

The authors go on to use the AEM classifier to identify senescence cells in FP-tagged A549 cells. The authors claim to have validated the classifier by comparing its result to GFP positive cells in their senolytic drug discovery setting (Figure 6). Next they perform as drug screen and use the GM classifier to assign drug treated cells to a senescent or non-senescent phenotype. They arbitrarily define a 40% threshold for hit detection and validate as upset of the hits using SA-beta-Galactosidase. They then perform a biological follow-up on already described senescence inducing drugs in cancer.

1. The classifier (Fig. 6e) only detects half as much senescent cells as the authors count GFP positive cells. This points towards a poor performance of the GM classifier (which isn't unsurprising given Fig. 5). Why did the authors decide to proceed with this classifier?
2. Also, the authors do not show classification results on the RFP positive cells. Please add a confusion matrix to this manuscript with RFP+ and GFP+ cells against senescent and non-senescent cells.
3. The analysis and QC of the drug screen is not sufficient (Fig 6e0). Aspects such as Drop outs, replicates (not displayed at all), and QC remain entirely unreported. It is unclear how the authors have ended up on a 40% cut off for hit detection. At this point it seems like an arbitrary value, enabling the exclusion of Dexorubicin from the hits (since it serves as a tool compound which induces senescence in both screened cell lines). The screening community has developed many analysis procedures for the proper analysis of drug screens, this manuscript does their efforts a disservice.
4. Why have the authors limited their follow-up analysis to I-BET726, which is accidentally the drug inducing the lowest senescence in A549. It would have been fairer to include multiple of these "negative control" drugs.
5. Some of the hits (blue box) are unlabeled and were not included in the follow up (Fig. 7). What are they and why were they not included?
6. In the yellow box only one drug is labeled, what are the other drugs called?
7. Has the screen identified any compounds previously unknown to induce senescence.

On the CSS/TSS

The authors design an scoring algorithm which integrates "single-cell" information to grade the level of senescence in a tissue. In a first step, cells of tissues are assigned to The algorithms hinges on the assignment of cells into senescence marker positive or negative cells, since . In a second step, the morphology feature values of the SM-negative cells are averaged, the SM-positive cells however, are first ranked in descending order based on SM intensity and the average of the morphology features using top 100 cells is then calculated. These two average measurements serve as reference phenotypes of senescent and non-senescent cells to which all cells per tissue are compared to using a sequence of arithmetic operations as defined by the authors. The result of this algorithm is the CSS. To my understanding (but I struggled to follow the description here), the authors then use the percentage of cells which had a CSS larger than 1 and smaller than 5 as the TSS.

The CSS and TSS seems chiefly dependant on SMs (p21 or p16 or others) for their calculation. The morphology features measured in the H&E section are used as proxy measurements to approximate the SM values in the adjacent section. The TSS can therefore only ever be as good as the SM in the detection of Senescence in a tissue. The value of the TSS over an simple SM score would be in its application on large collections of H&E tissue section images in order to make statements on Senescence on tissues which cannot be stained with an SM. To my understanding the authors do not show such "transfer" application. What I can find multiple times in the manuscript is what I would deem a control experiment, which is correlating TSS to SM levels. The fact that these correlations are low, does not bode well for the "transfer".

I have multiple comments and requests for clarification regarding the CSS/TSS.

1. Please describe in the manuscript the motivation for the development of the CSS/TSS and discuss its advantage over a simple SM-based Senescence scoring.
2. The authors have calculated the TSS for many tissues derived from many conditions, which is great. Please show that the TSS models generalise well by running the individual TSS models on other H&E tissue images without adapting parameters or the values of the phenotype references.
3. As described in the methods the authors defined multiple parameters to calculate the CSS/TSS, e.g. threshold for p21CIP1 positivity, AUC range of 1-5, and $a \geq 0.7$ (what is a btw?). It is not clear to me, how the authors reached these values. Please perform a robustness analysis for at least the two first parameters and show how dependent the correlation of TSS and the % of SM positive cells is. Also, please elaborate on whether these values were also used in the other tissue experiments and report this in the methods section.
4. The authors do not describe or show any quality control efforts of primary and derived data. Given the focus of the work this is surprising. I would kindly ask the authors to include a QC pipeline for all their tissue experiments, which at the minimum showcases their ability to exclude imaging artefacts in the tissue images, their ability to segment nuclei accurately, accurately, and their ability to exclude miss segmented nuclei. Providing the readers with such information is crucial to raise the confidence in the presented data and algorithm.
5. The design and execution of the CSS/TSS is a major methodological component of this work. I would ask the authors dedicate at least one panel of a main figure to describe the approach using a graphical summary.
6. Supplementary 8e contains a “Feature Impact correction” box, which is not mentioned in the Methods sections describing the CSS and TSS. What is this “Feature Impact correction”?
7. To my understanding the CSS/TSS is not a classifier nor is machine learning used in the core algorithm. Please adjust in the text and figure legends. Please correct me, should I be wrong.
8. I find the authors use of the term AUC confusing in this context (see methods). When talking about classifiers AUC would be used in conjunction with a ROC curve. The TSS/CSS isn't/ doesn't use classifiers. Please change the wording.

The TSS could be a great addition to the tool box of image-based characterisation of H&E tissues if and when its robustness and ability to transfer to unseen images is confirmed. Without these additional validations, the TSS appears to me like a complication to a problem that could be solved by staining with an SM of choice.

I have to admit that I struggle to fully understand the design choices made by the authors for the CSS/TSS algorithm. For now, this algorithm feels like a tailored solution with limited utility for others.

On the code

Additionally, the code used for this publication was not available under the provided link which hinders a thorough assessment of the work. While I understand that the authors might want to keep the training data private, I would strongly encourage them to deposit their code in a public repository.

Varia

1. Through the manuscript the authors report experiment means/medians, the data in Fig.1f) looks like distribution of single cell data. What test do the authors use to compare the two conditions? The tests mentioned in the methods section are not suitable here. Consider using for instance a Kolmogorov-Smirnov test, reporting both the p-value and the KS statistics.
2. Supplementary Figure 2b and c, Why are the ROC curves of the AEM classifier “edgy”?
3. Typo on page 8: “SA-b-Gal staining confirmed that MLN8054- and etoposide-treated, but not cells irradiated for 36 h were senescence (Fig 2n).” should say senescent.
4. I disagree with the conclusion statement on the bottom of page 8 and top of 9. “However, the fact that the AEM classifier identified MLN8054 cells (Fig 2o, in which we did not observe significant amounts of DNA damage) as senescent, shows that the classifier is not just detecting DNA damage but rather a combination of nuclear features associated with senescent cells.” The classifier doesn’t identify DNA damage nor senescence, but a morphological phenotype that is induced by a senescence inducing drug and partially by a cell cycle inhibitor.
5. On page 16 the authors mention “To determine if the CSS could predict senescence at the tissue level, we analysed the distribution of CSS scores in tissues with a relatively high (3.394%) or low (0.552%) percentage of p21Cip1-positive cells”. How did the authors get to the values to define high and low percentage?

Point-by-point rebuttal to reviewers' comments NCOMMS-23-29145-T

Reviewer #1

The authors have addressed my concerns. This is an interesting and potentially broadly useful study.

Thank you, we appreciate the feedback.

Reviewer #2

NCOMMS-23-29145-T

After reading the revised manuscript and the responses to my comments, I've noticed that the authors have effectively addressed the majority of my concerns. As a result, I firmly believe that the manuscript has significantly improved and is deserving of publication in Nature Communications.

However, there is one last point I would like to point out. It is widely known that, unlike other methods of inducing cellular senescence, when cellular senescence is induced by constitutively active MEK, cellular and nuclear morphology remain largely unchanged (DOI: 10.1101/gad.12.19.3008). Hence, it would be interesting to know if the classifier developed by the authors in this study can be applied to determine senescent cells in such cases. Therefore, it would be better for the authors to mention something on this point in the Discussion section.

That is a good observation, and we have addressed it in the text: 'In some instances, such as senescence induced by expression of constitutively activated MEK {Lin, 1998 #626}, the morphology of the nuclei is largely unchanged, so it is unlikely that our classifiers would work on those conditions, but our study defines a framework that could be adapted to identify most types of senescent cells as needed.'

Reviewer #3:

Summary

I have reviewed the work of Duran et al. with a specific focus on the data science, computer vision and result visualisation aspects presented in the paper. The work can be divided into two parts. In part 1, the authors develop a set of classifiers for the identification and assignment of tissue culture cells to a phenotype of choice (i.e. senescent, non-senescent). In part two, the authors develop a Cell and Tissue Senescence Score (CSS and TSS respectively), which they use to describe senescence in tissues.

The focus of the paper is on the methodology rather than the description of a biological discovery. Unfortunately, many key aspects of the Methods-type manuscript are underdeveloped here and would need to be included to justify publication in this journal. Further, I think that there are some technical mistakes which need addressing to ensure that the proposed method is truly adding value to the research community.

I applaud the authors for showcasing models for different types of primary data such as tissue culture cells and tissue sections and multiple application settings. However, I can't shake the feeling that this breadth of examples comes at the cost of quality and thoroughness. All parts of the manuscript have multiple, severe data science issues which raise quality concerns.

In its current form, I would not recommend the publication of this work. The authors would have to make very substantial changes to the overall structure and content of the manuscript to convince me that the data science in this work meets the standard Nature Communications readers have grown accustomed to.

On the structure of the manuscript: The manuscript is too long. I would encourage the authors to consolidate their findings into a total of 4 figures. This would help focus the message of the paper and limit redundancy and the potential for confusion. Specifically, I would suggest that the authors:

1. Try to emulate the structure of a methods-focused manuscript better. E.g. Introduce the new methods, show their methodological soundness, benchmark them against existing solutions, and show their utility in the discovery of biological results. Many excellent examples can be found in Nature Methods.

We appreciate the feedback from the reviewer on the format and length. While their points are sound, we have to balance the 'methodological' aspect of the manuscript with the questions that an audience of researchers interested in cellular senescence (such as the other reviewers) will have. We agree that incorporating experiments and suggestions made during the review process has affected the length of the manuscript and have tried to reorganise it and trim where we can (e.g. we have eliminated old Figure 4 as the data was presented in old Figure 5; move to the supplementary parts of Figure 2; deleted redundant parts of Figure 3; moved to supplementary parts of Figure 6 and 7). Still, we believe that the Nature Communication format (of a longer article) is appropriate for our article that is not just methodological but rather directed to a senescence audience.

2. Pick one classifier to highlight in the main manuscript. I appreciate the effort of trying multiple classifiers but a) it isn't surprising that multiple works b) it isn't clear to me what the value is of showing multiple classifiers in such depths, given that the authors only report results achieved with them rather than elaborating on the differences between the results generated by the classifiers. In essence, the authors call different functions of the sklearn python package. This isn't particularly innovative nor difficult.

What the reviewer suggests (to pick one classifier to highlight) is what we have mostly tried doing (with the AEM classifier). We have compared AEM with other classifiers to make the point that while they are classifiers that behave well in many instances, due to the heterogeneity of senescence, there might be a point to use a tailor-made classifier for specific uses (e.g. if we want to screen for drugs that cause senescence in a specific type of cancer such as glioblastoma, it might be better to build a specific classifier for glioblastoma following the methodology described here than relying on a universal senescence classifier).

3. For this flavour of publication, the methodology is overall poorly described in the main manuscript and particularly in the Methods section (what to describe is suggested further down in this document).

We have revised the methodology to include additional details. We have also consulted an expert in machine learning and AI to be sure that we cover all the details expected by researchers in that field. In addition, we have uploaded the code and underlying data to GitHub https://github.com/Sen-Lab-LMS/Senescence_nuclear_features.

We apologise to the reviewer as we had intended to do this in the previous submission but unfortunately, the private link we provided had expired and could not be accessed.

4. On the comparison of ML algorithms with human-curated features vs. Neuronal Networks

The authors discuss multiple times the pros and cons of classifiers versus Neuronal Networks. To my understanding, this comparison is particularly motivated by a recent paper using Neuronal Networks to identify senescence in tissue culture cells and senescence. I would advise the authors to stay clear of a comparison which isn't grounded in quantifications since either side has its benefits and shortcomings. I would expect a more balanced and critical discussion of their methodology, should the authors insist on including a pure qualitative comparison in their manuscript.

We have followed the reviewer's advice to stay clear of comparisons with the other manuscripts and describe them as a complementary approach: *'Neural networks were recently used to identify senescence from cell microscopy {Kusumoto, 2021 #6691;Heckenbach, 2022 #6755}. Given that the identification of senescent cells is a bottleneck in the field {Gil, 2023 #7236}, our work described here joins those as a complementary and much-needed approach.'*

On the classifiers presented in part 1

In the first part of the manuscript, the authors develop a suite of classifiers which use nuclear morphology features to assign tissue culture cells to two classes (senescent, non-senescent). The classifiers are used on data generated and extracted from multiple microscopes and image analysis software packages. In Figures 1, 2, and 3a) to 3i) the authors showcase a set of classifiers trained on perturbation data that results in mixed populations of senescent and non-senescent cells thus introducing class errors in the ground-truth data.

1.) The quality of the classifier immediately drops substantially when the authors apply the model in a test scenario. E.g. Figure 1h and 1i detecting ~90% of cells as senescent in train (AEM) and only 65% in test (AEM). While the authors are right that there is still a significant difference between DMSO and Eto treated populations in train and test, their performance drops by almost 30% (1-(65% / 90%). A similar performance drop can be seen in Figure 2 d) and e). This is normally a sign of a poor classifier. The authors don't report this performance drop properly in the result section nor do they touch on it in the discussion.

The issue that the reviewer picks here (with the test/training data shown in old Figure 1h-i) was not necessarily due to the poor performance of the classifier but to the fact that the test data on Figure 1i for SA- β -Gal staining and the classifier was not derived from the same samples but from samples run in parallel. The difference between the % of cells considered senescence (by SA- β -Gal or by the classifiers) mostly reflects differences in the degree of senescence induction between the samples. To avoid this issue, we have used a new test dataset in which SA- β -Gal and the classifiers were measured in the same samples (by using a fluorescent-based SA- β -Gal). The new data, directly comparable as it is measured in the same samples (new Figure 1i) do not show a drop in performance.

Moreover, I want to highlight that while that testing is done with test data in which we have maximal or minimal senescence, later on in the manuscript we address this more thoroughly by measuring senescence in ~80 samples in which we have a whole range of concentrations of senescent cells (as described in Sup Fig 3). This data allows us to infer correlations and precision, recall, accuracy and F1 for the algorithms (e.g. as shown in Figure 3i or Figure 4). This is a more demanding dataset and rigorous testing that more accurately reflects the performance of the classifiers.

2.) In Figures 1, and 2, the authors repeatedly compare the classifier results to the % of SA-B-Gal positive cells. It would improve the legibility of the figures and facilitate the comparison immensely if these results were displayed on the same Y axis. They recognise the limitation of these classifiers and implement a fluorescence marker-based class assignment of cells as ground truth. They then compare these improved classifiers to one of the best-performing original classifiers and argue that given the similarity in Precision, Recall, Accuracy and F1 scores, the original and improved classifiers are of the same quality (I am paraphrasing here). The authors then proceed to use the original classifiers for the analysis of a drug screen. The comparison between original and fluorescence-guided classifiers is too simplistic and potentially misleading.

As the reviewer explains, across figures 1-5, we compare the classifiers with SA- β -Gal positive cells as the 'ground truth'. Due to the intricacies of the senescence phenotype and the lack of universal senescence markers, we compare also the classifiers to senescence defined in other ways (e.g. p21+/BrdU_ or p21+/p53+ cells in Fig 3c-e).

To develop these original classifiers (described in Fig 1-2), we assume that all cells in the 'senescence-treatment condition' are senescence and all cells in the control condition have no senescence. To test whether that assumption was resulting in worse-performing classifiers, we generated new fluorescent-guided classifiers which we considered as senescence-only cells that we have confirmed as senescence using markers and vice versa (those algorithms are described in Fig 3j-l and ED Fig 3e-h).

A thing that might have been unclear in the previous submission and we have tried to clarify now is that (as the reviewer suggested at a later point), we compared the performance of the different classifiers **in the same datasets** (e.g. the table with precision, recall, accuracy and F1 shown in Fig 3l has been generated on the same dataset). That comparison shows a similar (if not better) performance of the AEM classifier when compared with the fluorescent-based classifiers. We believe that the direct comparison serves to make the point that our original classifiers perform similarly to these new ones. Similarly, comparisons presented in Figure 4 are on the same datasets (and directly comparable)

4.) In fact there is only one valid comparison setup to justify their conclusion of using the original classifiers interchangeably with the fluorescence-guided ones. Namely to use the fluorescence-assisted assignment of senescence should serve as ground truth. Thus, I would kindly ask the authors to define a set of cells from said ground truth apply

the “AEM” and the new classifier and evaluate the results using the previously mentioned metrics. This analysis good also be repeated multiple times to statistically test whether “AEM” is indeed as good as the “newer” classifiers. Pending the proposed analysis, no comparison should be drawn between the classifiers.

We apologise to the reviewer that it was not clear or well-explained that this is exactly what we were doing in old Figure 3l (new Figure 3i). We have added a new panel (Figure 3h) showing the set-up of the experiment to clarify. Since what we have already done is what the reviewer suggested was needed to compare the classifiers, we are confident that this issue should be resolved. Similarly, comparisons shown in Figure 4 are on the same datasets.

5.) The authors do not describe or show any quality control efforts of the primary and derived data. Given the focus of the work this is surprising. I would kindly ask the authors to include a QC pipeline for all their cell experiments, which at the minimum showcases their ability to exclude imaging artefacts in the images, their ability to segment nuclei accurately, and their ability to exclude miss segmented nuclei. Providing the readers with such information is crucial to raise the confidence in the presented data and algorithms.

To clarify the pipeline and quality control efforts, we have explained the QC efforts that we have carried out in methods (on the drug screen; high throughput microscopy image acquisition; on acquisition of images from slides).

6.) Many of the experiments are control experiments and should be presented in the Extended Data or Supplementary section of the work.

The expectations of a reader (or reviewer) interested in the methodology versus the senescence biology differ. Nevertheless, we agree, hate convoluted papers and have taken the point and restructured and organised the figures to clarify and improve the readability (as described in the general point 1 of this reviewer).

7.) The authors focus solely on morphology features. They could have included nuclear DAPI intensity and texture features (they are readily available in a CP pipeline). Could they elaborate on why they chose not to do this and/or show that the inclusion of such features doesn't improve the classifier performance?

From the data we have observed, and particularly with cancer cells, DAPI intensity is subject to inter-experimental variability that introduces noise. DAPI staining has experimental shortcomings to ensure a consistency that can offset the benefits of taking intensity into account. Similarly, texture data is linked to DAPI performance in image acquisition.

The authors then apply multiple classifiers on various pharmacologically perturbed and unperturbed cell lines (Figures 4 and 5).

8.) These are a great results, as the authors show the ability of the classifier to be used on unseen data, i.e. different cell lines.

We appreciate that the reviewer likes the result.

9.) However, it also shows the clear limits of a pre-trained classifier (Supplementary Figures 5b and 6). The authors have trained a classifier on A549 cells treated with Etoposide (AEM). When applied to A549 treated with another senescence-inducing drug (Barasertib) or on another cell line the performance drops substantially. To claim that “The above results suggest that senescence classifiers are generalizable to different stresses and cell types, beyond those used to train the algorithms.” Isn't well substantiated as there are striking indications that AEM does not generalise.

In the old Figure 4-5 (now summarised in Figure 4) we aimed to assess the performance of classifiers on unseen data (e.g. different cell types or senescence inducers) and compare how classifiers that have been trained differently compared. There are a few things to highlight from this analysis:

- As highlighted by the reviewer in their previous point, classifiers can predict 'unseen data' (e.g. detect senescence in cell types or caused by inducers others than those used for their training)
- However, (as highlighted by the reviewer on this point) given the heterogeneity of senescence, classifiers have limitations and they are not able to perform well in all types of senescence.
- Overall, we highlighted classifiers that are 'good enough' for many/most use cases, while highlighting that depending on the use case, to use/derive a specific classifier rather than relying on AEM or other of the more general classifiers described here can be the right choice.

In general terms, I think that we agree with this reviewers' assessment (expressed in points 8 and 9) and we have tried to rebalance the writing to express that correctly in the discussion: *'Overall, our classifiers identified many types of senescent cells with good accuracy, even those types not included in their respective training sets. However, given the heterogeneity of senescence, our classifiers performed worse in identifying some types of senescence (e.g. in IMR90 cells or in response to barasertib treatment). In some instances, such as senescence induced by expression of constitutively activated MEK {Lin, 1998 #626}, the morphology of the nuclei is largely unchanged, so it is unlikely that our classifiers would work on those conditions, but our study defines a framework that could be adapted to identify most types of senescent cells as needed.'*

10.) I do not understand the purpose and methodology of Figure 5. Why so many classifiers and how were they trained? I would kindly aske the authors to explain this experiment better. The authors use the GM model later in the manuscript. Maybe these tables could be supplementary.

As described above, the motivation for generating, describing and comparing the different classifiers (shown in old Figures 4 and 5, now summarised in Figure 4) is to be able to assess how differently trained classifiers compare (this allows us to produce the conclusions summarized in answer 9). We agree with the reviewer that this part of the manuscript was convoluted and have taken their suggestion of reorganising old Figures 4 and 5 into one Figure 4.

On drug discovery/ drug screen

The authors go on to use the AEM classifier to identify senescence cells in FP-tagged A549 cells. The authors claim to have validated the classifier by comparing its result to GFP-positive cells in their senolytic drug discovery setting (Figure 6). Next, they perform a drug screen and use the GM classifier to assign drug-treated cells to a senescent or non-senescent phenotype. They arbitrarily define a 40% threshold for hit detection and validate as upset of the hits using SA-beta-Galactosidase. They then perform a biological follow-up on already described senescence-inducing drugs in cancer.

1.) The classifier (Fig. 6e)) only detects half as much senescent cells as the authors count GFP-positive cells. This points towards poor performance of the GM classifier (which isn't unsurprising given Fig. 5). Why did the authors decide to proceed with this classifier?

We thank the reviewer for highlighting this discrepancy in the cell numbers. After revising the data analysis, we realised that there was a mistake in the way that the GFP and mCherry positive and negative populations were calculated (thresholding values to consider positive cells were not locked resulting in different cutoff values applied in the different replicates and conditions). This has been corrected and the new data has been presented in Figure 5e.

2.) Also, the authors do not show classification results on the RFP-positive cells. Please add a confusion matrix to this manuscript with RFP+ and GFP+ cells against senescent and non-senescent cells.

We have added a confusion matrix as asked by the reviewer in ED Figure 5d.

3.) The analysis and QC of the drug screen are not sufficient (Fig 6e0). Aspects such as dropouts, replicates (not displayed at all), and QC remain entirely unreported. It is unclear how the authors have ended up on a 40% cut-off for hit detection. At this point it seems like an arbitrary value, enabling the exclusion of Doxorubicin from the hits (since it serves as a tool compound which induces senescence in both screened cell lines). The screening community has developed many analysis procedures for the proper analysis of drug screens, this manuscript does their efforts a disservice.

While we understand the point of the reviewer, we politely disagree with their general assessment, as the screen was only aimed as a proof of principle and succeeded in identifying both already known and new drugs with the ability to induce senescence. Screens can be analysed in many ways, their nature is often exploratory and they need validation as it is often said, 'the proof is in the pudding'

Nevertheless, we take the reviewer's point and have revamped the analysis of the screen (shown in new Figure 5f-j). In the new analysis, we have calculated B-scores (as we have done in the past for other screens such as in Georgilis et al Cancer Cell 2018 or McHugh et al Nat Cell Biol in press. The criteria for hit selection, QC and else have been summarised in Figure 5g and explained in methods. Overall, we want to highlight that the results were similar to what we have described before.

4.) Why have the authors limited their follow-up analysis to I-BET726, which is accidentally the drug inducing the lowest senescence in A549? It would have been fairer to include multiple of these "negative control" drugs.

While the screen identifies drugs that selectively induce senescence in IMR90 or A549 cells (Figure 5j), the drugs that have therapeutic significance are the ones selectively inducing senescence in cancer but not normal cells, as those have the potential of being used as cancer therapy with reduced 'off-target' side effects, we concentrated on analysing that group of drugs. In the previous version of the manuscript, we included I-BET726 as an example of the drugs inducing senescence in normal (IMR90) but not cancer (A549) cells. It was there to show that the screen can also identify drugs like that, but since the therapeutic potential is on the drugs inducing senescence in cancer but not normal cells we concentrated on those. The reviewer had a point in spotting that the effects of I-BET726 were weaker, hence we swapped it for a different hit (AG-14361).

5.) Some of the hits (blue box) are unlabelled and were not included in the follow-up (Fig. 7). What are they and why were they not included? 6.) In the yellow box only one drug is labelled, what are the other drugs called? 7.) Has the screen identified any compounds previously unknown to induce senescence?

Following the guidance of the reviewer we have re-analysed the screening and that figure is now gone. Nevertheless, we are providing the source data of the screen including the value of all hits for whoever is interested in looking into the screen and the hits (or re-analysing the data). The screen has identified both drugs that were known to induce senescence (e.g.

niraparib, MLN8054, that therefore acted as internal controls) and drugs that were not known to induce senescence (e.g. ARQ621 or SC144, novel hits). We have highlighted that clearly in the text.

On the CSS/TSS

The authors design a scoring algorithm which integrates “single-cell” information to grade the level of senescence in a tissue. In a first step, cells of tissues are assigned to The algorithms hinges on the assignment of cells into senescence marker positive or negative cells, since. In a second step, the morphology feature values of the SM-negative cells are averaged, the SM-positive cells however, are first ranked in descending order based on SM intensity and the average of the morphology features using top 100 cells is then calculated. These two average measurements serve as reference phenotypes of senescent and non-senescent cells to which all cells per tissue are compared to using a sequence of arithmetic operations as defined by the authors. The result of this algorithm is the CSS. To my understanding (but I struggled to follow the description here), the authors then used the percentage of cells which had a CSS larger than 1 and smaller than 5 as the TSS.

The CSS and TSS seem chiefly dependent on SMs (p21 or p16 or others) for their calculation. The morphology features measured in the H&E section are used as proxy measurements to approximate the SM values in the adjacent section. The TSS can therefore only ever be as good as the SM in the detection of Senescence in a tissue. The value of the TSS over a simple SM score would be in its application on large collections of H&E tissue section images to make statements on Senescence on tissues which cannot be stained with an SM. To my understanding, the authors do not show such a “transfer” application. What I can find multiple times in the manuscript is what I would deem a control experiment, which correlates TSS to SM levels. The fact that these correlations are low, does not bode well for the “transfer”.

I have multiple comments and requests for clarification regarding the CSS/TSS.

1.) Please describe in the manuscript the motivation for the development of the CSS/TSS and discuss its advantage over a simple SM-based Senescence scoring.

Identifying senescence cells *in vivo* is a key bottleneck and pressing need of the senescence community. This is exemplified for example by the recent meeting about the subject held in Vienna to write a ‘minimal information’ white paper about how to identify senescent cells *in vivo* (currently under revision) or the comment piece that I have written for Nature Cell Biology about the problems to identify senescent cells (Gil et al Nat Cell Biology 2023). Some of the (many) problems related to detecting senescent cells are technical. Most of the samples for clinical research are stored as paraffin (that for example does not admit staining for SA-β-Gal that needs to be done in fresh cryosections). In addition, some of the antibodies used to stain for senescence will not work reliably in stored sections. Also, mutations in senescence genes will make it difficult to find reliable markers for some types of samples. The attraction of the CSS/TSS is that can produce a senescence score based on hematoxylin or hematoxylin & eosin staining, that will exist for almost all of the archival and new patient samples, without the need to worry about markers of senescence, staining and antibodies working. This might seem trivial for someone outside the field but it is a big bottleneck appreciated by those in the senescence field. We have tried to explain this more clearly in the text for non-experts on senescence.

2.) The authors have calculated the TSS for many tissues derived from many conditions, which is great. Please show that the TSS models generalise well by running the individual TSS

models on other H&E tissue images without adapting parameters or the values of the phenotype references.

We apologise if this part was not explained clearly. The evaluation of senescence on tissue sections using CSS/TSS (shown in Figures 7-9), is based **on a single TSS/CSS model** (described in Sup Fig 8). Therefore, the underlying question that this reviewer asks here (whether a TSS model has predictive value on other datasets without being modified) is already answered in the manuscript.

The model was trained with a subset of HDTV1 experiments in which RAS^{G12V} is expressed in the mouse liver (summarised in Sup Fig 8). Later we use that single model to:

- predict senescence in two different tests of experiments upon expression in mouse liver of RAS^{G12V} or RAS^{G12V, D38A} (Fig 7i-k) and Extended Data Fig 7e-f.
- assess the effect of a senolytic drug in the liver of mice (Fig 8a-d)
- assess senescence in a model of liver fibrosis in mice (Fig 8e-h)
- measure senescence in the liver of mice during ageing (Fig 8i-l)
- estimate senescence in patients with mild fatty liver disease (Fig 8).

3.) As described in the methods the authors defined multiple parameters to calculate the CSS/TSS, e.g. threshold for p21CIP1 positivity, AUC range of 1-5, and a ≥ 0.7 (what is a btw?). It is not clear to me, how the authors reached these values. Please perform a robustness analysis for at least the two first parameters and show how dependent the correlation of TSS and the % of SM-positive cells is. Also, please elaborate on whether these values were also used in the other tissue experiments and report this in the methods section. I will go step by step into the nature of these parameters:

- We set up a threshold of DAB=0.2 for considering cells p21 positive (see example of some cells with different p21 DAB intensities in Figure R1), the same that we have to set up a threshold for positivity of any of the other markers used in this story. This is based on both the distribution of the staining, the use of negative controls and the ability to distinguish background from signal.

Figure R1. **Representative pictures of cells stained for p21CIP1 and their relative DAB intensity values.** DAB>0.2 is the value chosen to consider cells positive. Scale bar, 10 μ m.

- AUC range of 1-5. We evaluated a range of AUCs (that we will from now on refer to as ‘% of cells with a CSS between 1-5’ to avoid confusion with using the term AUC for ROC), and determined that the % of positive cells with CSS values of 1-5 were the ones better defining the TSS score. We have incorporated a new Sup Table 4? in which we compare different ranges of values, which shows that despite other ranges performing well, 1-5 seems to be the most consistent across systems.
- The parameter ‘a’ measures circularity and in this context, we have used this parameter to exclude cell types other than hepatocytes in the liver sections of patients. This is not needed in the mouse samples (as size exclusion is sufficient there). These indications were already in the methods, but we have tried to clarify them further.
- We have analysed the dependency of the TSS and the CSS distribution curves of changes in two of the top parameters used to define CSS and TSS (Figure R2 for reviewers, below).

Figure R2. **Dependency of TSS and CSS of changes in nuclear area and nuclear perimeter.** Left, graphs show how a % change in the Area (a) or perimeter (b) affects TSS. Right, graphs show the changes in the cell senescence score curves if the original area (a) or original perimeter (b) changes 15%.

I hope that we have clarified the different points that the reviewer queries. We have better explained all of those details in the methods. Also, we have added Sup Table 4?, showing the correlation between the percentage of cells with CSS values at different ranges and p21 positive cells, which gives the rationale to pick up that value. If there are questions that we have misunderstood or missed, we can try to address them, but overall and more importantly, in answer to the last point: the values were defined in a training set (of oncogene-induced senescence in the liver) and **that single TSS/CSS model** (with all values fixed) has been used to evaluate senescence across other datasets and different conditions.

4.) The authors do not describe or show any quality control efforts of primary and derived data. Given the focus of the work, this is surprising. I would kindly ask the authors to include a QC pipeline for all their tissue experiments, which at the minimum showcases their ability to exclude imaging artefacts in the tissue images, their ability to segment nuclei accurately, accurately, and their ability to exclude miss segmented nuclei. Providing the readers with such information is crucial to raise confidence in the presented data and algorithm.

We have improved the description of the QC of this and other methodologies in the methods as described earlier.

5.) The design and execution of the CSS/TSS is a major methodological component of this work. I would ask the authors to dedicate at least one panel of a main figure to describe the approach using a graphical summary.

We have added a graphical summary in Figure 7d.

6.) Supplementary 8e contains a “Feature Impact correction” box, which is not mentioned in the Methods sections describing the CSS and TSS. What is this “Feature Impact correction”? We apologize for the lack of clarity, we were referring to the weighting of the different nuclear features as part of calculating the cell senescence score. In this revised version, we have explained with more clarity the approach taken (both in methods and in the figures) and as part of it we have eliminated the term ‘feature impact correction’ that was not clear and referred to in Sup Fig 8e to ‘*adjust for overall feature weight*’ that hopefully is more clear.

7.) To my understanding the CSS/TSS is not a classifier nor is machine learning used in the core algorithm. Please adjust the text and figure legends. Please correct me, should I be wrong.

Although TSS is at its core different to ML approaches used for in vitro studies, it is still essentially a classifier to the extent it classifies the cells based on the nature of parameters associated with each nuclei. Nevertheless, the reviewer has a point and for clarity we have referred to it as ‘*a score based in a combination of nuclear features*’. We are open to a better suggestion.

8.) I find the authors use of the term AUC confusing in this context (see methods). When talking about classifiers AUC would be used in conjunction with a ROC curve. The TSS/CSS isn’t/ doesn’t use classifiers. Please change the wording.

We used the term AUC as what we are doing is calculating the 'Area Under the Curve' (under the CSS distribution curves). However, we appreciate the point of the reviewer that 'AUC' is a standard term used in machine learning in conjunction with ROC curves. To avoid confusion, we have renamed it as '% of cells with CSS between 1-5', which is equivalent.

The TSS could be a great addition to the toolbox of image-based characterisation of H&E tissues if and when its robustness and ability to transfer to unseen images are confirmed. Without these additional validations, the TSS appears to me like a complication to a problem that could be solved by staining with an SM of choice. I have to admit that I struggle to fully understand the design choices made by the authors for the CSS/TSS algorithm. For now, this algorithm feels like a tailored solution with limited utility for others.

As stated above, we have incorporated new data to show the robustness of TSS. Also, we have clarified that once we develop the TSS classifier with training data on the Ras^{G12V} model (summarised in Sup Fig 8). Later we use that single model to:

- predict senescence in two different tests of experiments upon expression in mouse liver of RAS^{G12V} or RAS^{G12V, D38A} (Fig 7i-k) and Extended Data Fig 7e-f.
- assess the effect of a senolytic drug in the liver of mice (Fig 8a-d)
- assess senescence in a model of liver fibrosis in mice (Fig 8e-h)
- measure senescence in the liver of mice during ageing (Fig 8i-l)
- estimate senescence in patients with mild fatty liver disease (Fig 8).

Given the above points, combined with the inherent problems of detecting senescence *in vivo*, that usually is not as simple as staining with a senescence marker of choice (Gil et al. Nat Cell Biology 2023; DOI: 10.1038/s41556-023-01267-w), I hope the reviewer appreciates that the senescence community would be interested in this approach as part of their toolbox to measure senescence.

On the code

Additionally, the code used for this publication was not available under the provided link which hinders a thorough assessment of the work. While I understand that the authors might want to keep the training data private, I would strongly encourage them to deposit their code in a public repository.

We apologise that the code was not accessible to the reviewer. We deposited the code in GitHub: https://github.com/Sen-Lab-LMS/Senescence_nuclear_features

Varia

1.) Through the manuscript the authors report experiment means/medians, the data in Fig.1f) looks like the distribution of single-cell data. What test do the authors use to compare the two conditions? The tests mentioned in the methods section are not suitable here. Consider using for instance a Kolmogorov-Smirnov test, reporting both the p-value and the KS statistics.

We reanalysed the mentioned single-cell data based on K-S tests (Fig. 1f / ED Fig. 1a) and reported that as suggested by the reviewer. We have highlighted in the text that regardless of the significance, any of these individual nuclear parameters is a poor predictor for senescence per se, but that prediction power increases when combined in a classifier: *'All of these nuclear features, except form factor, were significantly different between senescent (etoposide-treated) and control (DMSO-treated) A549 cells (Figure 1f and ED Figure 1a).*

As none of these nuclear parameters alone could distinguish senescent cells from non-senescent cells, we used these parameters to devise machine-learning classifiers that could predict senescence.'

Our collaborators made an analogy with classical machine learning problems (like the first attempts at facial identification in the 1990s) in which aggregation of weak features increase the predictive strength.

2.) Supplementary Figure 2b and c, Why are the ROC curves of the AEM classifier “edgy”? The reviewer must be referring to the ROC curves presented in Sup Fig 2d and ED Fig 1d, this has been corrected. The ROC curve for the classification tree model contained an intermediate step to simplify the graph (matplotlib package), which resulted in its edgy shape, this has been amended to display the original values.

3.) Typo on page 8: “SA-b-Gal staining confirmed that MLN8054- and etoposide-treated, but not cells irradiated for 36 h were senescence (Fig 2n).” should say senescent. This has been corrected.

4.) I disagree with the conclusion statement at the bottom of page 8 and top of 9. “However, the fact that the AEM classifier identified MLN8054 cells (Fig 2o, in which we did not observe significant amounts of DNA damage) as senescent, shows that the classifier is not just detecting DNA damage but rather a combination of nuclear features associated with senescent cells.” The classifier doesn’t identify DNA damage nor senescence, but a morphological phenotype that is induced by a senescence-inducing drug and partially by a cell cycle inhibitor.

We have amended the text to reflect this point: ‘However, the fact that the AEM classifier identified MLN8054 cells (Fig 2l, in which we did not observe significant amounts of DNA damage) as senescent, shows that the classifier is not just detecting DNA damage-associated changes but rather a combination of nuclear features associated with senescent cells.’

5.) On page 16 the authors mention “To determine if the CSS could predict senescence at the tissue level, we analysed the distribution of CSS scores in tissues with a relatively high (3.394%) or low (0.552%) percentage of p21^{CIP1}-positive cells”. How did the authors get to the values to define high and low percentages?

We selected the samples with higher and lower levels of p21^{CIP1} positive cells from the dataset as a way to show how the curves change in the two ‘extremes’ of the dataset.

REVIEWERS' COMMENTS

Reviewer #3 (Remarks to the Author):

I thank the authors for their kind words and thoroughly addressing my comments.

The authors have responded to my main points of concern and they implemented changes to the manuscript that improve its clarity, readability and quality with regards to the presented and employed data science.

I very much welcome these improvements and recommend the publication of the manuscript in Nature Communications.